



# Exploring the interplay between state, structure and runoff behaviour of lower mesoscale catchments

Simon Paul Seibert[1], Conrad Jackisch[1], Uwe Ehret[1], Laurent Pfister[2], and Erwin Zehe[1]

[1]Karlsruhe Institute of Technology (KIT), Institute for Water and River Basin Management, Chair of Hydrology, Kaiserstrasse 12, 76131 Karlsruhe, Germany
[2]Luxembourg Institute of Science and Technology, Department Environmental Research and Innovation, Catchment and Eco-hydrology research group, 5 avenue des Hauts-Fourneaux, L-4362 Esch/Alzette, Luxembourg

*Correspondence to:* Simon P. Seibert (simon.seibert@kit.edu)

**Abstract.** The question of how catchments actually "function" has probably caused many sleepless nights as it is still an unsolved and challenging scientific question. Here, we approach this question from the similarity perspective. Instead of comparing single physiographic features of individual catchments we explore the interplay of state and structure on different runoff formation processes,

aiming to infer information on the underlying "functional" behaviour. Therefore, we treat catchments as lumped terrestrial filters and relate a set of different structure and storage descriptors to selected response measures. The key issue here is that we employ dimensionless quantities exclusively by normalizing the variable of interest by its limiting terrestrial or forcing characteristic. Specifically we distinguish extensive/additive and intensive/non-additive attributes through normalizing storage

volumes by maximum storage capacities and normalizing fluxes (e.g. discharge) by permeability estimators. Moreover, we propose the normalized temporal derivative of runoff as a suitable measure to detect intensity-triggered (high frequency) runoff production.

    Our dimensionless signatures evidently detect functional similarity among different sites for baseflow production, storm runoff production and the seasonal water balance. Particularly in the latter

case we show that normalized double and triple mass curves expose a typical shape with a regime shift that is clearly controlled by the onset and the end of the vegetation period which we can adequately characterize by a simple temperature index model. In line with this, temperature explained 70 % of the variability of the seasonal summer runoff coefficients in 22 catchments distributed along a strong physiographic and climatic gradient in the German part of the Danube basin. The proposed

non-additive response measure detected signals of high frequency intensity controlled runoff generation processes in two alpine settings. The approach, in fact edge filtering, evidently works when using "low-pass" filtered hourly rainfall-runoff data of mesoscale catchments ranging from 12 to 170 km$^2$.





We conclude that vegetation exerts a first order control on summer stream flow generation when the onset and termination of summer are more significantly defined by temperature than simply by the actual Gregorian day. We also provide evidence that properties describing gradients (e.g. surface topography) and resistances (e.g. hydraulic conductivities) may be much more powerful in explaining runoff response behaviour when they are treated as groups compared to their individual use. Lastly, we show that storage estimators such as the proposed normalized versions of pre-event discharge and antecedent moisture can be valuable predictors for event runoff coefficients: For some of our test regions they explain up to 70 % of their variability.

## 1 Introduction

### 1.1 Hydrological similarity as a weak form of causality

How close must two catchments be with respect to state and structure such that they produce runoff in a similar way? Based on the findings of Dooge (1986) we establish our studies essentially on the expectation that hydrological systems are essentially deterministic. Hence, identical inputs of energy and rainfall will cause an identical runoff response, if two identical catchments are in the same state. This crude deterministic paradigm is, however, of low practical use, because neither the system state nor the structural setup are exhaustively observable. Hence, we can at best postulate that similarity of structure and state of a terrestrial system implies "similar" functioning (Wagener et al., 2007; He et al., 2011; Zehe et al., 2014). While such a weak form of causality may be easily defined in qualitative terms, its translation into useful similarity measures for structure, state and runoff response is far from being a straight forward exercise, particularly at the lower mesoscale.

### 1.2 Challenges in defining structural similarity at the lower mesoscale

Parts of the confusion stem from the inherent equifinality and non-uniqueness of most of our governing equations (Beven, 1989; Zehe et al., 2014). This is particularly true for runoff because an integrated mass flux leaving a catchment control volume is a non-unique product of a driving potential gradient and the control volume conductance (or its inverse resistance). This implies that systems that largely differ with respect to the topographical controls on the driving gradients and the pedo-geological controls on the integral conductance may produce runoff in a similar fashion (Binley and Beven, 2003; Wienhöfer and Zehe, 2014). Topographic controls and pedological controls on runoff generation must thus be interpreted as group, to be able to judge how they jointly control runoff behaviour. This requires metric data sets on topography as well as on soil water and aquifer characteristics. While the former is available as highly resolved digital elevation and landuse maps, the latter can in most cases at best be estimated using (very) coarse soil and geological maps in combination with pedo-transfer functions - not to mention the absence of data characterising preferential pathways. It is, hence, no surprise that many model-based similarity studies rely on categorical soil





and landuse data and translate them into metric catchment descriptors by means of their areal share
(e.g., Merz and Blöschl, 2003; Hundecha and Bárdossy, 2004; Carrillo et al., 2011; Sawicz et al.,
2011; Ali et al., 2012; Kelleher et al., 2015). Notwithstanding that this approach is feasible when
representing similar catchments by similar parameters in conceptual models, it is way too simple to
be conclusive for similarity of runoff production in the real world.

### 1.3   Storage estimators and state estimators - how to normalize and how to achieve coherence?

Storage estimators such as antecedent precipitation (Heggen, 2001; Brocca et al., 2009), pre-event
discharge (Refsgaard, 1997; Graeff et al., 2012) or dynamic storage (Sayama et al., 2011) have
been shown to be helpful to characterise storage in the catchment control volume and the related
runoff proneness across scales (Tetzlaff et al., 2011). This is particularly appropriate when subsurface
storage capacity controls runoff production (Struthers and Sivapalan, 2007; Struthers et al., 2007a,
b), which implies runoff to monotonically increase with storage and thus to be limited by additive
quantities. Such quantities are rainfall depth and saturation deficit in the case of saturation excess
(Dunne and Black, 1970), or rainfall depth and subsurface storage in the case of subsurface storm
flow (Tromp-van Meerveld and McDonnell, 2006; Lehmann et al., 2006).

Additive storage measures can easily be derived from either the catchment water balance or ob-
served rainfall and discharge volumes (McNamara et al., 2011) and equally easily be upscaled, as soil
and aquifer water content are additive quantities. However, absolute storage is difficult to compare
between different pedolological settings as these measures require a meaningful normalization in
order to be related to runoff processes. Furthermore, dynamic storage (Sayama et al., 2011) depends
on the starting point of integration. As catchment inter-comparison studies should compare coherent
time series, this starting point needs to be carefully chosen to ensure that integration starts at the
same relate storage state. When the catchments of interest are spread across a wide topographic and
climatic range, the same Gregorian day might be a very inappropriate choice, as further elaborated
in section 2.1.

Notwithstanding the importance of storage estimators, they do not provide a full characteriza-
tion of the catchment state. The latter particularly requires information on where in the catchment
the water is stored (Nippgen et al., 2015) and whether it is subject to strong, weak or no capillary
and/ or osmotic forces. Unfortunately, soil water potentials, plant water potentials and piezometric
heads are intensive state variables and thus non-additive. They can neither be determined as residu-
als of a balance equation nor can they easily be scaled up in an additive manner (Zehe et al., 2006;
de Rooij, 2009, 2011). Hence, characterization of the full system state requires comprehensive, spa-
tially highly resolved data sets on both soil moisture and soil water potentials (Zehe et al., 2013). As
these are rarely available in mesoscale catchments, similarity and catchment inter-comparison stud-
ies are challenging to work with, giving a fairly incomplete characterisation of the system state. This





is particularly unpleasant because it is the potential gradients which determine the "forces" driving
water and energy fluxes (Kleidon, 2012; Zehe et al., 2014).

### 1.4 Dimensionless response measures for and beyond capacity controlled runoff formation

The striking success of similarity theory and scaling based on dimensionless quantities through-
out a range of disciplines such as hydraulics (e.g. Reynolds, Froude, Péclet number), acoustics
(e.g. Helmholtz number), chemistry (e.g. Dammköhler numbers) or micro meteorology (e.g. the
Monin–Obukhov length) motivated past research for useful dimensionless quantities characteriz-
ing hydrological similarity (e.g., Saghafian et al., 1995; Reggiani et al., 2000; Woods, 2003; Berne
et al., 2005; Struthers et al., 2007b; Woods, 2009; Schaefli et al., 2011). The Budyko curve (Budyko,
1961) is probably the most generally accepted dimensionless analysis technique in hydrology to as-
sess similarity in the steady state water balance by plotting the evaporative fraction against a dryness
index. In line with these studies we hypothesize that dimensionless state-response diagrams are suit-
able candidates for similarity assessment for catchment inter-comparison. Proper normalization of
state and response measures means to normalize using those climate and terrestrial system proper-
ties which limit runoff production. The rationale is that one can expect these dimensionless plots to
remain invariant, as long as the limiting factors remain unchanged.

Normalization in the case of capacity controlled runoff formation is straightforward as it is lim-
ited by additive quantities, essentially storage and rainfall volumes. We may hence treat catchments
as lumped terrestrial filters (Black, 1997) and normalize event scale runoff by total precipitation
amount, and relate this to storage estimators such as antecedent precipitation (Heggen, 2001; Blume
et al., 2007; Graeff et al., 2012), dynamic storage (Sayama et al., 2011) or pre-event discharge (Gra-
eff et al., 2009; Kirchner, 2009; Zehe et al., 2010). A feasible normalization of storage estimators
should be based on the minimum and maximum subsurface storage volume/depth or, if this infor-
mation is not available, on the storage depth in the root zone of the soil. Similarly, we may compare
annual double mass curves of normalized accumulated rainfall and runoff fluxes to discriminate dif-
ferences in the seasonal interplay of storage and release (Pfister et al., 2002; Hellebrand et al., 2008).
Although all these measures and their normalization can in principle be determined as residuals of
the water balance and from available maps, the devil lies in the details as further elaborated in section
2.1.

Detection and normalization of intensity controlled runoff production is, however, not that straight
forward (Struthers and Sivapalan, 2007). Intensity controlled runoff generation is characterized by
intensive, convective rainfall forcing and a fast, high frequency stream flow response, reflecting on-
set of rapid subsurface flows (Lehmann et al., 2006; Wienhöfer and Zehe, 2014) and/ or infiltration
excess (Niehoff et al., 2002; Zehe et al., 2005). The latter is difficult to observe *in situ* during natural
forcing conditions but its occurrence is well known from many artificial rainfall simulation experi-
ments (e.g., Fiener et al., 2011, 2013). Intensity controlled runoff production occurs in a threshold





like manner (Lehmann et al., 2006; Zehe and Blöschl, 2004; Struthers and Sivapalan, 2007; Zehe
et al., 2007; Ruiz-Villanueva et al., 2012) and is neither controlled (and limited) by additive rainfall
properties nor by current storage. Hortonian overland flow production is for instance controlled (and
limited) by the relationship of non-additive rainfall intensity and soil infiltrability (Horton, 1939;
Hohmann, 2014). The latter is a conglomerate of unsaturated hydraulic conductivity, and suction

head as well as of the density, depth and capacity of apparent macropores (Beven and Germann,
2013). Again, none of these quantities is additive during up-scaling.

As intensity control implies i) the high frequencies to be dominant and ii) first order control of
non-additive characteristics, any form of spatial and temporal data aggregation essentially implies
to loose parts or even the complete signal due to low-pass filtering. There are promising options to

assess highly resolved patterns of rainfall based on weather radar (e.g., Ehret et al., 2008; Kneis and
Heistermann, 2009) or to estimate catchment scale patterns of biotic macropores (Palm et al., 2013;
Van Schaik et al., 2014). However, discharge as our best observation of runoff formation inevitably
represents a convolution of distributed runoff production and concentration, which inherently implies
low-pass filtering.

A cardinal question is thus on the minimum requirements for detecting intensity controlled runoff
generation. Related studies often operate at relatively small scales, relying on high frequency rainfall-
runoff data in combination with breakthrough or flushing of either contaminants (Gassmann et al.,
2013), artificial tracers (Wienhöfer et al., 2009), sediments (Martínez-Carreras et al., 2010) or even
diatoms as *smart* tracers (Martínez-Carreras et al., 2015; Klaus et al., 2015). Most "operational" data

sets however do not offer these sources of extra information and are at best available at an hourly
resolution and for catchment sizes $\geq$ 40-50 $km^2$. The challenge to detect intensity controlled runoff
production within inter-comparison studies seems at first sight similar to the challenge to repair a
watch with a monkey wrench. One way forward might be to relate temporal changes in rainfall in-
tensities to temporal changes in runoff - which means in fact to analyse the acceleration of input and

output fluxes, as further elaborated in section 4.4.

### 1.5 Objectives and research questions

While being fully aware of all the listed challenges and shortcomings of operationally available data
sets, we propose and test dimensionless measures to discriminate differences in runoff generation
(storage and/ or intensity controlled) in lower mesoscale catchments. In particular, we pose three

main questions:

– Question 1: How feasible is the use of dimensionless state and/or storage-response diagrams to
detect differences in event scale flood production, baseflow generation and the seasonal water
balance?

– Question 2: Can we detect intensity controlled runoff formation as essentially a high frequency
process based on low frequency data?

– Question 3: Which structural, climatic and ecological catchment characteristics explain the
differences between different catchments and among different years and which of them operate
in groups?

Our study area is the Bavarian part of the Danube basin in Southern Germany, which we introduce
in detail in section 3 together with the data and model we use. More specifically, we use an opera-
tional data set from the federal water resources management agency and standard categorial data on
landscape characteristics in about 130 lower mesoscale catchments. Additionally, we apply a cali-
brated water budget model which covers all of the sub-basins to ensure a consistent estimation of
evapotranspiration and storage estimators such as dynamic storage. Particular emphasis within our
inter-comparison study is on the issues of i) proper normalization of storage estimators and fluxes,
ii) assuring coherence and similar quality of associated time series and iii) on an assessment of the
different storage estimators with respect to explanatory power and redundancy.

## 2   Conceptual framework and candidate diagnostics

In this section we propose a set of dimensionless "functional diagnostics", suitable for catchment
inter-comparison studies across a wide range of end members. Hence, we exclusively rely on com-
monly available landscape properties and hydro-meteorological data.

### 2.1   Requirements of functional diagnostics

Useful diagnostics for runoff response and catchment state need to be sensitive to the limiting factors
and allow for a normalization of the responses and state variables to i) separate meteorological from
terrestrial controls and ii) to test our perception on underlying structural controls. We expect intensity
controlled runoff production to occur in landscapes characterized by strong gradients, shallow and
poorly developed soils, high abundances of either very coarse or fine/clay substrates, sparse surface
coverage, and/or geologies which develop rift aquifers. Capacity controlled runoff generation is
deemed to dominate in landscapes characterized by weak gradients, well drained and homogeneous
textured soils (without remarkable clay or skeleton contents), and medium to high degrees of surface
cover over parent materials that sustain pore aquifers. Also snow dominated areas are expected to
exhibit capacity controlled behaviour. At the seasonal scale we additionally need to make sure of
comparing coherent time series of similar data quality.

#### 2.1.1   Normalization of states and response measures

In our study we compare three storage estimates: a rescaled version of dynamic storage, accumu-
lated antecedent precipitation and pre-event discharge (see details in section 2.2). All three surrogates





yield estimates of absolute storage depths (L). Their normalization requires measures for storage capacity of the different catchment subsurface compartments, which should reflect both total and active storage volumes as well as the fractions of free water and capillary bounded water in soil. Estimation of these storage properties is hampered by the unknown depth of the lower boundary of the control volume and the heterogeneity of the subsurface materials (Troch et al., 2003; Spence, 2007; Soulsby et al., 2009). We thus normalize the storage estimators using average root zone field and air capacity, which are available through national soil maps (BGR, 1995). Despite their limitation in the vertical direction, these estimates are deemed to provide an indication of the relative importance of the storage volume containing capillary bounded water, which feeds evaporation and transpiration, and free water feeding groundwater recharge and runoff production (Zehe et al., 2014).

Normalization of rainfall-runoff response and seasonally accumulated runoff is straightforward in the case of capacity controlled runoff production by means of either total rainfall depth of an event or total annual precipitation. Baseflow during dry spells (radiation driven conditions) requires a different normalization based on estimates of aquifer permeability/transmissivity as these control water release. If this information is not available, as in our case, the average soil hydraulic conductivity provides an alternative.

### 2.1.2 Coherence and quality of integral storage measures

Estimators of water storage such as dynamic storage $(dS)$ (Sayama et al., 2011) depend essentially on the starting point of integration (Pfister et al., 2003). Coherence, in terms of "achieving comparability", of storage time series hence requires that integration in all catchments starts at the same relative storage amount. This could for instance be after significant dry and/or wet periods, when subsurface wetness can be deemed as being either near saturation or near the minimum. Particularly in the case of a strongly seasonal climate, distinct dry and wet periods can be useful in selecting a proper start date.

As $dS$ is i) based on the assumption of a closed water balance and ii) calculated from (areal) estimates of precipitation and model based estimates of evapotranspiration, related uncertainties have a direct effect on the storage estimator. A straight forward quality check of $dS$ is to plot it against normalized accumulated precipitation for several years, using the long term annual mean precipitation for normalization. By comparing patterns of $dS$ for time periods of potentially similar accumulated input one may detect trends, non-monotonic step changes or other inconsistencies. In our case most of the 130 datasets did not pass this benchmark test (compare section 3).

Finally, we face a similar challenge of "when to start" when relating integral storage measures to normalized baseflow. This is because "onset" and "duration" of the baseflow recession may have variable definitions and meanings (Blume et al., 2007). Furthermore, discharge at the river gauge is an aggregation of runoff production, concentration and routing along the river network. These





processes cover different spatio-temporal scales which make it increasingly difficult to determine a direct relationship of baseflow behaviour to integral storage measures when moving up in scale.

## 2.2 Candidate storage, response and intensity estimators for baseflow, runoff events and the seasonal water balance

This section introduces normalized storage and runoff response measures, their combination into dimensionless storage/state-response diagrams as well as their statistical analysis. We distinguish among i) the generation of baseflow during radiation driven conditions, ii) rainfall-runoff events as the driven case and iii) the seasonal water balance. The latter is separated into the winter term and the vegetation period, to explore the impact of vegetation controls. Our candidate diagnostic measures for high frequency runoff processes and intensity control are introduced at the end.

### 2.2.1 Normalized storage measures

Firstly, we use a normalized and re-scaled version of dynamic storage ($dS^*$) (see Appendix A1 on this aspect). $dS^*$ is calculated as the residual of the water balance equation, using estimates of areal precipitation ($P$), model based estimates of evapotranspiration ($E$) and observed discharge ($Q$). As given in Equation 1 we use the average soil storage volume for normalization, characterised by the sum of effective field capacity ($eFC$) and air capacity ($AC$) in the root zone ($\tau$), since metric information on aquifer capacity is not available. Estimates of $eFC_\tau$ and $AC_\tau$ are taken from the national soil map of Germany (BGR, 1995):

$$dS^*(t) = \frac{\sum_{t=1}^{T} P(t) - Q(t) - E(t)}{AC_\tau + eFC_\tau} \tag{1}$$

$dS^*$ is deemed to represent the total active bulk catchment water storage and we expected it to be associated mainly with deeper storage compartments and hence to control the slower flow processes. Values of $dS^*$ around zero indicate dry conditions whereas values near one indicate that dynamic storage is equal to the root zone storage volume. Note that both values > 1 (e.g. during the occurrence of snow) and < 0 may occur and absolute values must not be interpreted.

The second storage estimator is chosen to better reflect near surface storage. Similar to other studies (Heggen, 2001; Brocca et al., 2009; Graeff et al., 2012) we estimate normalized antecedent moisture ($\theta^*$), which is equal to the difference between precipitation and evaporation totals within the last seven days (T=7 days in Equation 2) normalized again by the average soil storage volume:

$$\theta^*(t) = \frac{\sum_{t=T-7d}^{T} (P(t) - E(t))}{AC_\tau + eFC_\tau} \tag{2}$$

Lastly we use a normalized specific pre-event discharge ($Q^*$) averaged across the last seven days:

$$Q^*(t) = \frac{\frac{1}{n} \sum_{t=T-7d}^{T} Q(t) dt}{AC_\tau + eFC_\tau} \tag{3}$$





The main disadvantage of $Q^*$ is that it cannot be attributed to any specific subsurface storage
compartment as it inevitably represents a combination of both, storage and release. The advantage is
that it relies on the best observation we have.

### 2.2.2 Baseflow generation during non-driven conditions

To explore controls of catchment structure and storage on baseflow generation we relate specific
baseflow depths ($Q_b$), normalized by the bulk average catchment hydraulic conductivity, to the dif-
ferent storage measures. To this end we define baseflow conditions as follows: ET < 0.1 mm, no
occurrence of snow, $\frac{dQ}{dt} < 0$ and no input in $P > 0.1 mm$ for a period of at least one, three and
five days. The one- and three-day period data sets are used to visually inspect how fast Q decreases
after precipitation ceases, which indicates how fast the terrestrial filter properties become domi-
nant. Within our statistical analysis we exclusively consider stream flow data where the last input
in $P > 0.1\,mm$ was at least five days ago. For response normalization we use the arithmetic av-
erage saturated hydraulic conductivity ($Ks$) of the catchment (Equation 4), since other estimators
for bedrock permeability were not available. $Ks$ is estimated for each catchment based on available
grain size distribution using *Rosetta's* pedo-transfer functions (Schaap et al., 2001).

$$Q_b^* = \frac{Q_b}{Ks} \qquad (4)$$

Normalized baseflow is then related to $dS^*$ and $\theta^*$ using the Spearman's rank correlation coef-
ficient ($\rho$) and the non-parametric test of significance proposed by Best and Roberts (1975). In the
case of significant relations (p-values<0.001), we try to identify an empirical storage-baseflow rela-
tionship by fitting power laws using $dS^*$ and $\theta^*$ as predictors using *R* (R Development Core Team,
2015). The quality of these relationships are judged by comparing their root-mean-squared-error to
the standard deviation of the normalized baseflow values (nRMSE).

Our approach is in line with past attempts to relate stream flow variations and drainage behaviour
of hillslopes or catchments (e.g., Laurenson, 1964; Rodríguez-Iturbe and Valdés, 1979; Brutsaert and
Nieber, 1977) and searches for feasible storage-baseflow relationships (Kirchner, 2009). Here, we
also test the spatial consistency of these storage-discharge relationships by comparing the multiplier
in the power law (as estimate of the effective catchment permeability) to the corresponding variation
of $Ks$ between the catchments.

### 2.2.3 Event scale rainfall-runoff response

At the scale of individual rainfall-runoff events we relate event runoff coefficients ($CR_E$), defined
as total event quick flow volume ($\sum Q_E$) divided by total precipitation ($\sum P_E$) (Equation 5), to
the different storage measures. To assure comparability of runoff coefficients, as recommended by
Blume et al. (2007), and to assure a sufficiently large sample we use an automated detection of
rainfall-runoff events based on a modification of the constant-k method (Blume et al., 2007) (details





on the method are provided in A2).

$$CR_E = \frac{\sum Q_E}{\sum P_E} \qquad (5)$$

We then select rainfall-runoff events with daily precipitation depth $\geq 10$ mm and calculate both, coefficients of determination (Pearson) and Spearman rank correlation coefficients among $CR_{Es}$ and the three different normalized storage estimators. Significant relationships are identified by p-values<0.001 in a two-sided t-test or the non-parametric test of Best and Roberts (1975). These are interpreted as evidence for capacity controlled runoff production. In case the respective storage

measure were uncorrelated, we test multiple regressions between $CR_E$ and $dS^*$, $Q^*$ and/or $\theta^*$, respectively.

### 2.2.4   Storage control on seasonal runoff generation

To shed light on the seasonal dynamics of catchment storage and release we compile normalized annual double mass curves (nDMC) and triple mass curves (nTMC) for different hydrolog-

ical years. Normalized double mass curves relate cumulated runoff ($cum.Q/\sum P$) to cumulated precipitation ($cum.P/\sum P$). The normalized triple mass curve adds cumulated evapotranspiration ($cum.E/\sum P$) as the third dimension to the plot. The rationale is to check whether the annual water balance is closed within a hydrological year, or whether the system carries stored water into the next year. The winter period and vegetation periods are separated using a temperature index model

proposed by Menzel et al. (2003) and analysed separately.

    The nDMCs within different catchments are compared according to a) the average and mean absolute deviations of their slopes within the winter and vegetation period, b) the presence and onset of a regime shift marked by plateaus and c) the mean and inter-annual variation of the annual runoff coefficient ($CR_{yr}$). Regime shifts are further analysed based on the anti-correlation of summer and

winter runoff coefficients ($CR_S$ and $CR_W$) with actual annual evaporation from available water balance simulations. Finally, we attribute differences within the double and triple mass curves to a range of different (n=24) structural and climatic properties of the catchment including temperature sums, characteristics of the grain size distribution, surface cover and several others. Particularly, we test the product of topographic gradient and saturated hydraulic conductivity as an explanatory

variable, as they are considered to act in concert.

### 2.2.5   Intensity controlled runoff generation

Our initial idea was to detect high precipitation rates as those being larger than the estimated hydraulic conductivity and to compare this to peak flow of events normalized with peak intensity of rainfall. To correct for the temporal mismatch between the maxima $P$ and $Q$, we intended to employ

a mean response time defined on the lag cross correlation between $P$ and $Q$ for each individual event (Kirchner, 2009). However, this approach did not yield clear signals due to several likely reasons.





Although hot spots in rainfall intensities are known to be localised and dynamic (Goodrich et al., 1995; Fiener and Auerswald, 2009), we are left having to treat them as spatial rather uniform values due to the low density of rain gauges in our study area. Moreover, texture based estimators using the Rosetta pedo-transfer functions (Schaap et al., 2001) remained as the only option and left us without a proper estimator of the influence of preferential pathways.

To separate high intensive rain showers from low and moderate intensive events we next calculate normalized rainfall event duration ($T_E^*$ (h)) as the ratio of total event rain depth ($\sum P_E$) divided by the maximum observed precipitation intensity ($P_{E,max}$), for all rainfall events exceeding a threshold of 10 mm. The threshold of $10\ \mathrm{mm\ h^{-1}}$ is recommended by the German Weather Service (DWD) to detect strong rainfall events.

$$T_E^* = \frac{\sum P_E}{P_{E,max}} \tag{6}$$

We expect convective, high intensive and extreme rainfall events to cluster at short normalized event durations with a large total amount. Consequently, we relate the maxima in the temporal changes of discharge ($dQ_{E,max}$) and precipitation ($dP_{E,max}$) (both in $\mathrm{mm\ h^{-2}}$) - which implies relating the acceleration of rainfall with stream flow mass. As high frequency processes are characterised by sharp peaks, we expect this normalized and dimensionless intensity change ($I_E^*$) (Equation 7) to separate intensity controlled from capacity controlled runoff production as intensity controlled conditions cluster at large $I_E^*$. Note that $I_E^*$ is an intensity measure and thus non-additive. It is independent from the runoff coefficient.

$$I_E^* = \frac{dQ_{E,max}}{dP_{E,max}} \tag{7}$$

Both, the normalized event duration and the normalized maximum change in stream flow are jointly analysed within scatterplots. Here, we expect intensity controlled processes to cluster around small values of $T_E^*$ and large values of $I_E^*$. Additionally, we compile three-dimensional scatterplots using $\sum P$, $dP_{E,max}$ and $dQ_{E,max}$ on the x, y and z-axis respectively. Here we expect high $dQ_{E,max}$ to be associated with high $dP_{E,max}$ whereas $\sum P$ is deemed to be unimportant.

## 3   Study area and dataset

The feasibility of the above introduced signatures is tested by inter-comparing operational data from the Bavarian Environmental Agency (LfU) for 130 catchments located in the Bavarian Danube basin ($\approx 45.000\ km^2$). In section 3.2 we detail the differences in the climate and physiographic setting of our test catchments and present our perception of the dominant hydrological processes. Before that, we will briefly discuss the quality of the database, which in fact was in most catchments so poor that the majority of the sites had to be excluded from the analysis.



### 3.1 Data quality and selection of headwater catchments

We focus on lower mesoscale catchments to minimize routing effects and select all gauged headwater sites $\leq 170 \, \mathrm{km}^2$ within the Bavarian part of the Danube basin. For this, hourly hydro-meteorological time series from the period 01.11.1999 until 31.10.2004 are available. The data base in the resulting 130 catchments is analysed according to a set of different quality criteria. We only include catchments where i) at least one meteorological station was closer than 20 km, ii) the total absolute water

balance error was smaller than 5 % , iii) the amount of missing and/or implausible meteorological data was < 5 %, and iv) where the streams are not subject to any severe regulation. This screening resulted in only 22 catchments being classified as suitable for the analysis. The sites are spread across the Bavarian part of the Danube basin (Fig. 1 and Appendix A3).

– Figure 1: MAP STUDY SITE –

The densest coverage in meteorological stations was for precipitation with a total number of 244 stations. The coverage of the other meteorological variables was much coarser, with 59, 55 and 43 stations for temperature, humidity and radiation, respectively. Since these numbers include stations which are up to 20 km apart from the finally selected 22 headwater catchments, we even tolerated

lower densities in meteorological stations as e.g. in the Mopex data set (Schaake et al., 2000; Duan et al., 2006). The lowest densities of meteorological stations are located in the southern alpine areas and the corresponding foothills and in the north-eastern parts of the Bavarian Danube catchment. Catchment characteristics were derived from different digital map products of Germany such as for soil (scale: 1:1,000,000) (BGR, 1995), hydrogeology (scale: 1:1,500,000 and 1:500,000) (Duscher

et al., 2015; BGR and SGD, 2015) and geology (scale: 1:1,000,000) (Toloczyki et al., 2006), a digital elevation model with resolution of 25 m, the CORINE land use data (as of 2006) and the official stream network provided by LfU. Last but not least, we employed the conceptual hydrological model LARSIM (Ludwig and Bremicker, 2006) as it provides consistent areal estimates of evaporation, rainfall and snow water equivalent. LARSIM has been calibrated for all study catchments and

operates at hourly time steps. Evaporation is simulated using the Penman-Monteith equation. Model input is based on interpolated station data (grid point method, US Department of Commerce, 1972). Additional information on the Bavarian part of the Danube basin, the hydro-meteorological data and the Larsim model can be obtained from Seibert et al. (2014).

### 3.2 Landscape setting and perceptual models of runoff generation

Topography, landuse, geology, soil and aquifer properties are highly variable among the different headwaters of the Bavarian part of the Danube basin, as the region was unequally covered by ice during the last ice age. The remaining 22 catchments reflect the entire physiographic range (see Tables 1 and 2). This is underpinned by the large range of topographic gradients ($\phi$) (Table 1) calcu-





lated according to McGuire et al. (2005) as the flow path length from each pixel to the stream divided
by the corresponding difference in height using the Whitebox geographic analysis toolbox (Lindsay
J.B., 2014). The climate gradient is also rather strong with mean annual precipitation ($MAP$) rang-
ing from 600 mm in the northern sub-catchments to more than 2000 mm in the southern, alpine areas
and a total average of 1000 mm. Annual potential evapotranspiration ranges from 350 to 600 mm.
Both $P$ and $E$ regimes are characterized by distinct seasonal cycles. In some of the southern alpine
areas more than 50 % of the precipitation may fall as snow.

Based on the dominant physiographic properties (referencing to Table 1 and 2) we grouped the
22 catchments into 5 major classes, which largely rely on (hydro)geology and detail on the expected
dominant processes drawing from Peschke et al. (1999) and Schmocker-Fackel et al. (2007), which
are categorized into being capacity controlled or intensity controlled:

- The "Alpine sites" (ALP) in the very south are dominated by poorly developed and shallow
  soils (average root zone depth $\leq$ 35 cm) with high contents of skeleton and coarse material
  (average pore volume 110 mm, $Ks \approx 1e - 6\,m\,s^{-1}$) over highly productive fissured (partly
  karstified) aquifers. The surface cover is sparse with a clear dominance of forests and mead-
  ows. Rock outcrops occur, particularly above the tree line which is approximately around
  1800 m.a.s.l in this environment. Catchments of this physiographic region (ALP1 ... ALP4,
  n=4) exhibit strong geopotential gradients (median $\phi = 0.36$) and receive about 1500 mm an-
  nual rainfall. These characteristics clearly suggest a dominance of rapid flow paths i.e. surface
  runoff, pipe flow/by-passing and rapid sub-surface stormflow. Here, we might thus expect high
  frequency, intensity controlled runoff formation, at least during extreme conditions.

- The "Triassic catchments" (TRI) (n=3) are composed of well-drained, poor to moderately
  developed sandy soils (mean root zone depths of 50-80 cm) with high portions of coarse
  material. Regosols, rendzinas, cambisols and partly podzols with rather weak vertical differ-
  entiation over calcareous sandstone (TRI1, TRI2) and sandstone (TRI3) prevail. The parent
  material sustains moderately to highly productive pore and fissured aquifers. The land use is
  dominated by arable land (about 60 %). Long-term mean annual precipitation is around 750
  to 800 mm. The median gradients within the three catchments differed slightly between 0.028
  and 0.038 (-) which is an order of magnitude smaller than in the alpine areas. These charac-
  teristics suggest a perceptual model, where subsurface matrix flow dominates, and the aquifer
  strongly controls runoff generation. However, coarse substrates and corresponding structures
  may also sustain rapid flow paths and saturation excess during high intensity rainfall events.

- The faulted "molasses basin" (MOL) and adjacent transition areas belong to a heterogeneous
  region which hosts seven of our sites on mostly well developed, medium and deep cambisols
  (root zone depths > 70 cm) with high contents of aeolian sediments (silt and loess). The parent
  material is often composed of sheet gravel (MOL1, MOL2, MOL3, MOL5) and sedimentary



rock and fluvial sediments (MOL4, MOL6, MOL7) which predominantly sustain low to mod-
        erately productive aquifers. In these catchments pore volumes are partly well above > 300 mm.
        The soils are fertile which promotes an intensive agricultural use. The surface topography is
        characterized by soft hills and U-shaped valleys. The corresponding gradients in geopotential
        are weak. Therefore, we expect that sub-surface capacity controlled (matrix) flow is the dom-
inant runoff process. However, during high intensity rainfall Hortonian overland flow (due to
        surface crusting on arable land) and saturation overland flow due to reduced hydraulic con-
        ductivity is deemed to create a mixture of capacity and intensity controlled runoff formation.

    –   The catchments in the "Bavarian Forest" (BFO) (n=3) consist of loamy, partly sandy cambisols
        with comparably high contents of skeleton (in some areas up to 75 Vol.-%). These lie over
crystalline granite and gneiss which are fractured but practically non-aquiferous rocks. The
        root zone depth is on average 60 cm. Forests and meadows cover 60 to 90 % of the surface.
        The topography is more pronounced and median gradients reach values up to 0.08. In these
        areas we expect that preferential flow pathways contribute significantly to runoff generation,
        but merely in a capacity controlled manner.

–   The data set also includes four catchments from the "Alpine Foreland" (AFO1…AFO4). Like
        the MOL-area this region exhibits complex characteristics as it was altered by three different
        glacial advances (and retreats). Consequently, we observe high spatial variations in the geo-
        logical parent material and thus, also in the soils, land use and hydrological characteristics (see
        Table and 1 and 2). The same applies for topography, as it is a relict of the different glacial
periods. The relief, though composed of similar gradients as e.g. the Triassic or Molasse sites,
        includes a rich variety of landforms typically found on ground, end and lateral moraines such
        as rolling foothills, (glacial) lakes, swamps and smaller surface water courses. Hence, there is
        no single dominating perceptual model on runoff formation available. Also the importance of
        different storage compartments cannot be estimated for this region as a whole, but needs to be
evaluated individually for each site.

        To further illustrate variations in the seasonal water balance, we present regime curves for four
    different catchments (Fig. 2). The catchment TRI1 (Fig. 2, top left) receives a fairly constant input
    in $P$ throughout the year, but releases $Q$ with a strong seasonality and pronounced minimum during
    summer. Compared to the other sites the inter-annual variation in $E$ is rather large. The catchment
AFO4 (Fig. 2, top right) in contrast shows seasonality in $P$ but a fairly constant output in $Q$. ALP4
    (Fig. 2, bottom left) and ALP2 (Fig. 2, bottom right), which are both alpine sites, show a pronounced
    minimum in $Q$ during February due to snow storage. ALP2 however shows a very large range in both
    $P$ and $Q$ during summer, which suggests little buffering and a high reactivity. In contrast, ALP4 has
    a much more damped response to $P$ during summer and a more pronounced seasonality in $ET$.

– FIGURE 2 KERNEL DENSITY ESTIMATES –





## 4 Results

The following section documents the performance of our diagnostics. More specifically, we present selected dimensionless storage/-state-response analyses that corroborate either their feasibility or their failure in discriminating differences in runoff behaviour in combination with the selected sta-
tistical measures introduced in section 2.2.

### 4.1 Storage and structure control on baseflow generation

During low flow conditions, the storage estimators $dS^*$ and $\theta^*$ are in most cases linearly independent. Table 3 presents the corresponding statistics. However, significant relationships are encountered in 7 out of 22 cases, with the highest Spearman rank correlation coefficient ($\rho$) between $dS^*$
and $Q_b^*$ is 0.39 with an average of 0.07. All catchments except one have high and significant rank coefficients of determination between $dS^*$ and $Q_b^*$ with values ranging from 0.12 to 0.88 and an average of 0.59. Hence, $dS^*$ seems to be a valuable predictor for low flow in a rather wide range of environmental conditions. We also find significant relationships between $\theta^*$ and $Q_b^*$ in about 50 % of all cases but with rank correlation coefficients being all smaller than 0.28 (except for catchment
AFO2, were $\rho$ was 0.54). Hence, $\theta^*$ possesses much less predictive power.

– Table 3: STATISTICS OF NON-DRIVEN CONDITIONS–

Five catchments (MOL5, MOL4, TRI1, MOL1, BFO3, MOL7) reveal power model exponents close to $1 \pm 0.3$ for their estimated normalized storage-discharge relationship (Table 3). This corroborates a linear storage-baseflow relationship in line with Fenicia et al. (2006). For the remaining
catchments, we obtain exponents clearly different from 1, suggesting a non-linear interplay of storage and baseflow production in the majority of the catchments. This finding is also supported by 2D scatterplots (Fig. 3) which clearly show a strongly non-linear relationship between $dS^*$ and $Q_b^*$ at e.g. AFO3, MOL6, MOL2, BFO1, JUR1 and other sites. The nRMSE of the estimated storage-baseflow relations was on average 0.74, with best values of 0.43 and worst values larger than 1.
Hence on average, the storage-discharge relationships are a better predictor than the average $Q_b^*$. Furthermore, we do not find a distinct spatial pattern in the exponents as both cases (linear and non-linear) occurred throughout different geologies and climate settings.

The normalized baseflow is in fact the flow multiplied with the inverse of the conductance. Due to the gradient-flux relationhip we thus expect the multipliers in the normalized storage discharge
relationship to partly reflect the strength of the gradient driving baseflow production. In line with this we find that the median of the catchment gradients explains ($r^2 = 0.28$, $p = 0.023$) of the multipliers when leaving out two alpine sites.

– FIGURE 3: EMPIRICAL STORAGE-BASEFLOW RELATIONSHIP –





### 4.2 Storage control on rainfall-runoff response

The total number of rainfall events within the four year period (n=14854) ranges from 334 to 859 among catchments. Omitting those influenced by snow or triggered by rainfall totals smaller than 10 mm yields 1174 rainfall events (53 events per catchment on average). The distribution of the corresponding $CR_E$ is right skewed with a median $CR_E$ of 0.06, a mean of 0.11 and a maximum value of 0.83 (inter-quartile range = 0.13).

Table 4 presents the resulting statistics of the analysis of the driven case. Significant rank correlations among the $CR_E$ and the three storage measures $dS^*$, $Q^*$ and $\theta^*$ are found in 9, 15 and 8 out of 22 cases, respectively. The corresponding average $\rho$ values are 0.61, 0.63 and 0.52, respectively. In all cases where $dS^*$ is significantly correlated to $CR_E$, $Q^*$ is significantly correlated to $CR_E$ as well. Basically the same applies for the comparison of $Q^*$ and $\theta^*$. Whenever $\theta^*$ is significantly

correlated to $CR_E$, $Q^*$ is significantly correlated to $CR_E$ as well, except for the catchments ALP1 and ALP4. We may thus state that $Q^*$ is the best predictor for $CR_E$ with both the highest number of significant cases and the highest $\rho$ values.

– Table 4 STORAGE STATISTICS OF DRIVEN CONDITIONS–

Furthermore, we find a rather interesting regional pattern where a distinct storage measure per-
forms much better. $Q^*$ is consistently not (significantly) correlated to $CR_E$ in the four alpine sites while $\theta^*$ has significant $\rho$ values in three of the four catchments. Consistently with this, plots of $CR_E$ versus $\theta^*$, thereby scaling the point size with rainfall depth, clearly corroborate the dominant influence of rainfall depth (and probably intensity) in the Alpine catchments (e.g. ALP4, Fig. 4, top left) (see also section 4.4). A remarkable finding from the Triassic catchments is that the three
catchments TRI1, TRI2 and TRI3 have the highest $\rho$ and $r^2$ values between $CR_E$ and $Q^*$ among the entire data set, with up to 70 % of explained variance (compare Table 4). Relationships between $CR_E$ and $\theta^*$ are often pretty linear here, whereas that between $CR_E$ and $dS^*$ indicates threshold behaviour (see Fig. 4, top right). Also the catchments located in the Molasse area often show significant $\rho$ values between $CR_E$ and $Q^*$ and the functional relationship is linear in most cases (e.g.
MOL5, Fig. 4, bottom left). However, the relationships are clearly more noisy than in the Triassic area. The catchments located in the Bavarian forest have rather similar $\rho$ values for both, $dS^*$ and $Q^*$ and the functional relationship appears linear as well (see e.g. BFO3, Fig. 4, bottom right).

– FIGURE 4: STORAGE CONTROLS ON EVENT RUNOFF COEFFICIENTS (4x1) –

The inter-comparison of the storage measures at the beginning of the rainfall-runoff events re-
veals $dS^*$ and $\theta^*$ as being not significantly correlated, except for one catchment (MOL5) (Table 4). However, $Q^*$ is significantly correlated to $dS^*$ in 14 and to $\theta^*$ in 12 out of 22 cases, although the catchments do not coincide. The statistics also reveal that multiple linear regressions among $CR_E$ and the two most explanatory and uncorrelated storage measures explain - at best - 5 % of additional





variance ($r^2$) in all except three catchments (ALP3, ALP2, MOL7) compared to the univariate re-
gressions. Hence, we conclude that the consideration of different uncorrelated storage measures
does not further improve the predictability of the event runoff coefficients compared to the single
best predictor.

Similar to the non-driven case we tested for significant relationships between average and median
$CR_E$ and $Ks$ times $\phi$. The resulting coefficients of determination are very small in all cases ($r^2 \leq$
0.02). Also the three regression slopes between $CR_E$ and the different storage measures obtain small
and insignificant coefficients of determination. However, it turned out that the median topographic
gradient alone explains 31.4 % (p-value=0.0066) of the variance of the average $CR_E$ between the
catchments. We may state that gradient and resistance are conjunct during baseflow recession when
the system operates close to local equilibrium conditions. During rainfall conditions the gradients
dominate the concert, which indicates further-from-equilibrium conditions.

### 4.3 Seasonal interplay of storage and release

#### 4.3.1 Normalized double mass curves

The double mass curves are similar in all catchments in terms of a fairly linear increase in the winter
period and a clear regime shift towards much flatter, partly zero slopes in the vegetation period.
In fact the slopes of the nDMCs are almost constant, just parallely shifted, during the period of
vegetation at many sites (e.g. at MOL2, TRI3 or BFO1, Fig. 5, top left, top right and bottom right,
respectively). Strikingly, the onset of the vegetation period, defined by a temperature index model
(Menzel et al., 2003) accurately predicts when the regime shift occurs (in terms of $cum.P/\sum P$).
Moreover, temperature sums explain 70 % of the variance of the summer runoff coefficients with
respect to the entire range of our physiographic setting. During winter, temperature aggregates are
not significant and without predictive power (Fig. 5, bottom right). We may thus state that the onset
of the vegetation period dominates the seasonal interplay in storage and release during the "summer"
period, in all of our physiographic and climatic settings (except for the alpine region).

– FIGURE 5: nDMC AND SCATTERPLOT CR vs. TEMPERATURE SUMS –

The different catchments within our data set show considerable variations in the seasonal summer
and winter runoff coefficients and partly also with respect to their inter-annual variation (Table 5).
On average the seasonal winter runoff coefficients ($CR_W = 0.67$) exceed the average summer runoff
coefficients ($CR_S = 0.32$) by a factor of 2, with two exceptions (ALP2 and ALP3, both alpine sites).
The mean absolute deviation (mad) of the seasonal runoff coefficients are twice as large during
winter ($mad_{CR_W} = 0.1$) as during summer ($mad_{CR_S} = 0.06$) (Table 5).

– TABLE 5: STATISTICS AND SLOPES OF THE nDMCs –





With respect to the different physiographic settings we encounter distinct seasonal and spatial patterns. During winter the highest average nDMC slopes ($CR_W = 0.8 - 0.9$) occur in the north eastern catchments (BFO1, BFO2, BFO3) which are rather densely forested, but also in the alpine ALP1 catchment. ALP4, ALP3 and ALP2, which are also alpine catchments and located on similar altitudes, show much lower winter runoff coefficients of 0.64 to 0.71 on average, probably due to storage in the snow pack. The smallest winter runoff coefficients (0.35 and 0.55) occur in MOL7 and TRI3, respectively. With respect to the inter-annual winter variance we encounter small mean absolute deviations $\leq 0.05$ in low lying sites of the Molasse and glacial drift areas e.g. MOL5, AFO4, AFO2, AFO1. High mean absolute deviations $\geq 0.15$ occur in different geologies, including the sites TRI2, BFO1, JUR1 and BFO3. Please also note that $CR_{yr} > CR_W$ in a few cases where snow exhibits a strong control on winter runoff regimes (e.g. ALP3 or ALP2, see Table 5). Here, fitting linear regressions to the double mass curves is not suitable for estimating seasonal winter runoff coefficients.

According to the statistical analyses in which we regressed 24 different variables against the slopes of the seasonal winter runoff coefficients, the most explanatory variables are sand content ($r^2 = 0.29$), median gradient ($\phi$) times $Ks$ ($r^2 = 0.22$), silt content ($r^2 = 0.22$), forest coverage ($r^2 = 0.16$), skeleton content ($r^2 = 0.15$), number of frost days ($r^2 = 0.14$), effective field capacity ($r^2 = 0.13$) and absolute sum of negative temperatures ($r^2 = 0.12$). All other variables have coefficients of determination $r^2 \leq 0.10$. In several multiple linear regressions based on the above mentioned variables the best result is achieved for a combination of $\phi$ times $Ks$, forest cover and absolute sum of negative temperatures (multiple $r^2 = 0.30$, p-value<0.001). Active storage estimates ($dS$), summer temperature sums and length or end of the period of vegetation from the previous hydrological year do not help to improve the prediction of the actual $CR_W$. The key finding in this analysis is that $\phi$ times $Ks$ yields a $r^2 = 0.22$, whereas two variables alone only explain 0.02 and 0.08 % of the variance in the $CR_W$, respectively. This corroborates that surrogates for gradients and resistances act jointly and that their impact is detectable even at the lower mesocale.

The summer season is characterised by an opposite spatial pattern compared to the seasonal winter runoff coefficients. The highest seasonal $CR_S \geq 0.8$ is found in the snow-dominated alpine catchments of ALP3 and ALP2. The smallest $CR_S$ with values between 0.07 and 0.12 are encountered at the Triassic sites (TRI3, TRI2, TRI1). It is also important to note here that several low-lying sites the $CR_S$ shows very little inter-annual variance as indicated by mean absolute deviations $\leq 0.03$ (e.g. MOL5, TRI3, TRI2, MOL6, MOL2, MOL4, JUR1, MOL3 and others) (Table 5). In these catchments the slopes of the nDMCs are fairly constant throughout different hydrological years indicating a very strong control of evapotransiration on the water balance during summer. At these sites the curves of the nDMCs in summer have nearly identical slopes and are simply shifted in parallel depending on the onset of vegetation activity.





We may hence state that normalized double mass analyses are powerful tools for discriminating seasonal differences in the interplay of storage and release among mesoscale catchments. However, they do not provide insights into the reasons for inter-annual variations. In several cases we observed inter-annual variations in $CR_{yr} \geq 0.1$, which could stem from variations of $P$ or $ET$ or a carry over of water storage into the next year. To provide more insights we introduce normalized triple mass curves by adding $cum.E/\sum P$ as a third dimension.

### 4.3.2 Normalized triple masse curves

Conceptually we usually assume that the change in storage tends to zero within a single hydrological year. Hence, we assume that large inter-annual variations in the rainfall-runoff ratio $CR_{YR}$ coincide with large inter-annual variations in the evapotranspiration ratio $(CE_{YR})$. To evaluate this assumption on our data set we construct normalized triple mass curves and calculate the mean absolute deviation for both, $CR_{YR}$ and $CE_{YR}$. Within our sample we find several catchments where the mean absolute deviation in the evapotranspiration ratio $(mad_{CE_{yr}})$ is rather similar to the mean absolute deviation in the annual runoff coefficients $(mad_{CR_{yr}})$ e.g. MOL5, MOL2, ALP1, MOL4, or MOL1 (see examples in Fig. 6, upper row). However, we also find several sites where $mad_{CR_{yr}}$ clearly exceeded $mad_{CE_{yr}}$ e.g. TRI3, TRI2, BFO1, JUR1, BFO3 or BFO2 (compare Fig. 6, lower row). This may be attributed to a carry over of water storage feeding runoff formation (blue water) between the hydrological years, indicating inter-annual memory (under the assumptions of a closed control volume). Only in two catchments (AFO4 and MOL7) $mad_{CR_{yr}}$ is substantially smaller than the corresponding $mad_{CE_{yr}}$. This can be explained by a carry over of water into neighbouring years, feeding $ET$ (green water).

– FIGURE 6: , nTMCs: ALP1, MOL2, TRI2, BFO1 –

## 4.4 Intensity controlled runoff formation

### 4.4.1 Data evidence in Alpine catchments

Strikingly, we also find signatures of intensity controlled runoff in two Alpine catchments (ALP1 and ALP2). This is illustrated in Fig. (7) which compares two flood events from site ALP2 caused by rather similar totals of rainfall (244 and 200 mm) and identical event runoff coefficients ($CR_E = 0.58$). The rainfall intensities as well as the discharge peaks in the right panel are however twice as large compared to the left panel, yielding considerable differences in the normalized temporal intensity changes ($I_E^* = 0.08$ vs. $I_E^* = 0.32$). Similar rainfall-runoff dynamics with strong temporal changes in $P$ which are followed by strong increases in $Q$ are observed during many events at site ALP1.

– Figure 7: OBSERVED EVENTS –

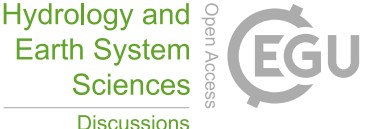



In these catchments the highest normalized temporal intensity changes indeed cluster at small normalized event durations (Fig. 8, left panel). The same scatterplot for MOL2 (Fig. 8, right panel) reveals normalized temporal intensity changes to spread equally across all event durations. The three-dimensional scatterplots of normalized temporal runoff changes against total precipitation and max-

imum intensity (Fig. 8, lower row), reveal clearly that large runoff changes coincide partly with high intensities and small rainfall totals. We may, hence, state the proposed signatures are feasible for detecting high frequency runoff even within low frequency data sets in mesoscale catchments. To illustrate that this is of more than academic importance we present a comparative model exercise.

– Figure 8: INTENSITY MEASURES –

**4.4.2 Explorative modelling for intensity limited runoff formation**

We compare two different model concepts to further elaborate the feasibility of our diagnostics for detecting intensity control. Our comparison shall particularly highlight the errors we might expect when simulating intensity controlled runoff formation with models relying on capacity controlled runoff formation with respect to the events depicted in Fig. 7. Specifically, we compare the HBV

beta store (Bergstroem, 1976) with a Green and Ampt approach (G&A) using the solution of Peschke (1985), as typical concepts for capacity and intensity controlled runoff formation. Both runoff generation concepts are implemented in R (R Development Core Team, 2015) and combined with a simple linear reservoir, whereas surface runoff is allowed to bypass the latter in the case of G&A. Both implementations are then fitted to observed stream flow data in an event based mode. Here we optimize

the maximum storage depth ($SMax$), beta parameter ($\beta$) and the reservoir constant ($k_{res}$) in the case of HBV using a simulated annealing algorithm in combination with the root-mean-squared-error as objective function. The G&A approach is parametrized based upon a Rosetta Schaap et al. (2001) estimate of $Ks$ and a literature value for the suction head ($psi$) at the wetting front (Maidment, 1993). The parameters of the linear reservoir ($SMax$ and $k_{res}$) are adopted from the HBV optimization to

ensure identical conditions.

During both events depicted in Figure 9 the HBV type setup outperforms G&A, when being judged on the Nash-Sutcliffe-Efficiency (NASH) criterion. During intensity controlled conditions the HBV bucket concept however clearly fails to reproduce the high runoff frequencies in terms of the slope of the rising limb and in the peak discharge (compare black box in Fig. 9, right panel). A

closer look at the right panel reveals, however, that G&A matches the magnitude of peak discharge (which is important for flood warning) much better than the beta store model. The slightly worse NASH value is because the timing error in peak occurrence is punished, which is a well known deficiency of the NASH statistical analysis (Seibert and Ehret).

– Figure 9) modelling of intensity controlled processes –





This exercise suggests that we are much better off in capturing sharp peaks of high frequency, intensity controlled runoff formation processes when using model concepts that are sensitive to the controlling, intensive properties compared to model concepts which do not account for this issue. Though this is essentially not new, data driven diagnostics which assist in deciding whether intensity controlled processes need to be considered in hydrological modelling are novel and rarely available.

**5    Discussion and conclusions**

In this study we propose various dimensionless diagnostics to characterize differences in terrestrial runoff production of catchments at the seasonal and the event scale. Particular emphasis is on a) their suitable normalization and b) on the question whether low-passed rainfall-runoff data from mesoscale catchments still bear detectable signals of high frequency, intensity controlled runoff pro-
duction. As benchmark we use operational rainfall-runoff data from 22 catchments spread across a wide range of physiographic and climate conditions in the Bavarian part of the Danube catchment.

**5.1    Normalized double mass curves discriminating seasonal runoff behaviour**

Normalized double mass curves turn out to be an easy-to-compute, yet very powerful means to detect similarity and differences in the seasonal water balance. In our case their general shape is
(invariantly) characterized by a linear increase in the winter period and a regime shift with small near to zero slopes when vegetation starts to control the water balance. The onset and duration of this regime shift could be predicted very well by a simple temperature index model (Menzel et al., 2003). In line with this, temperature explains 70 % of the variability of the summer runoff coefficients within the 22 catchments. It is noteworthy that the usual (Gregorian) definition of spring
and fall onset are of little help to predict the regime shift here. We hence conclude that vegetation exerts first-order control on stream flow generation in "summer", while onset and end of the summer (i.e. the vegetation period) is defined by temperature conditions rather than simply by the Gregorian day. This finding is important as it suggests that phenological data (and corresponding surrogates) provide valuable information which is mostly not included in standard hydrological data (or at least
hardly considered). We further conclude that any assessment of the "pure abiotic controls" of the catchment water balance should be restricted to (snow free) periods of the dormant season.

The variability of winter runoff coefficients is generally much less predictable by the available structural and climatic descriptors. Also the different storage estimators are of little use. The most interesting finding is that the rather coarse estimates of the catchment soil hydraulic conductivity
and the median gradient operate indeed as a group and that their impact is even detectable at lower mesoscale sites: their product explains 22 % of the variance in the winter slopes, while either of the values itself is an insignificant predictor. Expressing runoff by the product of an effective gradient





and a control volume conductance, requires the system to operate close to local thermodynamic
equilibrium conditions. We hence conclude that this is at least partly the case at the seasonal scale.

Also normalization of the double mass curve is straight forward. Here we use annual precipitation
totals of the respective hydrological year. The advantage is that both axes are normalized to one.
The disadvantage is that the same amount of relative accumulated rainfall does not correspond to
the same total rainfall amount. This would require us to normalize precipitation and discharge with
the long term mean annual precipitation at the prize, that the maximum ordinates would then not be

constrained to one.

We also provide evidence that normalized triple curves are well suited for explaining inter-annual
variability of annual runoff coefficients either by different accumulated evaporation totals or by carry
over in storage. The drawback of this signature is that it needs a calibrated water balance model for
its calculation, due to the required estimates of $E$, while the double mass curve relies exclusively on

standard observables.

### 5.2   Pre-event discharge as best predictor for capacity controlled runoff production

At the event scale we compare plots of event runoff coefficients against the three partly independent
storage estimators i) normalized dynamic storage $dS^*$, ii) accumulated normalized discharge $Q^*$
and iii) the normalized difference between antecedent precipitation and evaporation, $\theta^*$. Generally

$Q^*$, though it is the less sophisticated measure, does clearly outperform the other storage measures
in terms of the explained variances and with respect to the number of catchments with significant
rank correlations. Yet the comparison of different storage measures reveals regionally specific domi-
nances. Particularly for the alpine catchments, $Q^*$ is insignificant while antecedent wetness explains
most of the variance within three out of four alpine catchments. This is in line with our perception

that these basins are composed of shallow soils and we thus expect a higher importance of the near
surface storage.

Another interesting finding is that the median topographic gradient explains 31 % of the variability
of the mean catchment event runoff coefficient averaged over all 22 catchments. In fact we find
the same result for the 90 % quantiles of the runoff coefficients and when using the square root

of the topographic gradient. We hence conclude that this correlation reflects fairly well the strong
dependence of rainfall totals and intensity on elevation (and the gradient) rather than the influence of
the topographic gradient as a force for driving runoff concentration during rainfall driven conditions.

In comparison to $Q^*$ and $\theta^*$ and with regard to the effort of its derivation, normalized dynamic
storage provides little additional value. Thereupon, $dS^*$ is significantly correlated to $Q^*$ in 14 out

of 22 catchments, although it is to be expected that parts of these correlations are spurious (Pear-
son, 1897; Kenney, 1982), as both, $Q^*$ and $dS^*$ are calculated upon the same variable. Similarly,
we argue that correlations between $dS^*$ and response measures such as event runoff coefficients
are more difficult to evaluate as they may also involve (non-trivial) spurious fractions because both





variables are again calculated based on discharge and precipitation. Partly, this applies for $Q^*$ and
$\theta^*$ as well, but certainly to a lower degree. We hence conclude that $dS^*$ is of limited use for explaining rainfall driven runoff formation compared to the well known pre-event discharge and the well known antecedent precipitation. We furthermore conclude that these different storage measures characterise storage in different depths, and that event runoff production is controlled by different storage compartments in different regions.

Our results corroborate that the event runoff coefficient is a useful and easy to calculate normalized response measure to discriminate capacity controlled runoff formation. However, it fails to discriminate high frequency runoff production processes as underlined by our findings in the alpine catchments ALP1 and ALP2. We also conclude that normalization of different storage estimators is generally helpful to compare their sensitivity ranges, even when relying on such simple estimates as the root zone storage depth or the pore volume therein, as done here.

### 5.3 Heterogeneous performance of storage-baseflow relations

The occurrence of 19 (out of 22) significant relations between $dS^*$ and $Q_b^*$ confirms that $dS^*$ is a meaningful storage measure for the prediction of low flow conditions under a range of different (humid) physiographic settings. We also found specific storage-baseflow relationships by fitting power laws, with their exponents and factors being sensitive to changes in the physiographic setting. This finding is in line with the results of Shaw and Riha (2012) who derived storage-discharge relationships for several catchments of up to 6400 $km^2$ in size using an adaptation of the method proposed by Kirchner (2009).

A closer look reveals however that the estimated storage-baseflow relations are only partly of convincing quality. In many places they are rather noisy despite the fact that the nRMSE suggests predictive power. This corroborates that visual inspection of such relations is indispensable before using them for instance to parametrize regional baseflow production in a model. Parts of the noise in the relations most likely arise from the inherent data uncertainty. However, normalization of $dS^*$ by the root zone storage volume and of $Q_b$ by $Ks$ is also error prone. $Ks$ is at best a surrogate for the aquifer conductance and thickness. The soil map (BGR, 1995) suggests that e.g. BFO1 and TRI1 have identical $Ks$ values (according to Rosetta estimates (Schaap et al., 2001)). However, the hydrogeological map (Duscher et al., 2015) reveals that the aquifer underlying BFO1 is composed of virtually all non-aquiferous fissured rock, whereas the subsurface of TRI1 hosts low to moderately productive pore aquifers (compare Table 2). The alpine sites show similar contradictions as the smallest $Ks$ values coincide with the highest productive aquifer types (still Table 2). A more meaningful structural normalization of $Q_b^*$ would thus require estimates of aquifer transmissivity. We hence conclude that the identification of catchments with similar baseflow production was not feasible with the proposed approach.





Within our attempts to explain differences in the power law multiplier based on catchment char-
acteristics, we derive $\phi$ based upon surface topography and not, as ideally required, upon bedrock
topography. Nevertheless, we find that a considerable portion of the variability is explained by the
topographic gradient, in line with the flux–gradient–conductance relation. Given the large number
of significant relationships between $Q_b^*$ and $dS^*$ we conclude that $dS^*$ is a feasible predictor for
baseflow production within a rather wide range of physiographic settings. This supports our initial
assumption that $dS^*$ characterises deep storage.

**5.4  "Edge filtering" of low-passed data to detect high frequency runoff processes**

The proposed intensity signature detects evidence for high frequency intensity controlled runoff gen-
eration in two alpine catchments within the available low frequency data sets. The key is to "edge-
filter" both rainfall and discharge data by taking their temporal derivatives and then to normalize the
maximum runoff change by the maximum in precipitation change. This response measure separates
the available rainfall-runoff events into subsamples with high normalized temporal intensity changes
clustering at small event durations. We conclude that the approach we present is an easy-to-apply
technique to test for the occurrence of intensity controlled runoff generation processes. This is also
relevant for hydrological modelling as we show that a wrong conceptualization of intensity con-
trolled runoff by a capacity controlled model approach might imply that the model misses the flood
peak (even though it has a good NASH statistic). Thus, data driven signatures on high frequency
intensity controlled runoff generation can assist in the conceptualization of hydrological models or
serve as structural benchmarks. We hence conclude that high frequency runoff production might
play a much more prominent role in lower mesocale flood production than we usually conclude
from analysing hourly (or even lower) resolution data sets. Therefore, we recommend that opera-
tional data should be recorded and stored with at least 5 min resolution, as this might reveal high
frequency processes operating even more frequently than we expect.

**5.5  Conclusion and Outlook**

Overall, we recommend the following signatures as suitable to discriminate differences in terrestrial
runoff production in lower-mesoscale catchments:

- Normalized double mass curves for the seasonal water balance,

- the event runoff coefficient in relation to pre-event discharge for capacity controlled runoff
  formation,

- the event duration in combination with the normalized intensity change for detecting high
frequency processes.

The onset and termination of the vegetation period are useful to explain differences in the summer
water balance. We also argue that gradients and conductances - and hence their underlying controls





- are not independent if one attempts to explain functional differences by differences in the physiographic and climate setting. However, despite the good explanation we found for differences in the summer water balance, we were not able to robustly link functional similarity to structural similarity, based upon the properties available within our (operational) data set.

This brings us to our last conclusion which is founded on the dilemma of quantity vs. quality. On the one hand, inter-comparison studies require large sample sizes to include a sufficient number of end-members and to avoid type I errors (false hits). This makes widely available operational data sets indispensable (at least for the moment). On the other hand, we need accurate and sufficiently resolved data beyond rainfall and runoff to avoid type II errors (false negatives). Such data are (at least for the moment) only included in "research" data sets which are fairly limited in number and spatial distribution. To increase confidence in the proposed signatures we suggest they be applied to i) to a larger number of catchments and ii) to a (nested) set of small and densely instrumented catchments with homogeneous geological setup.

## Appendix A

### A1  Re-scaling and consistency of integrative storage measures

To ensure that two catchments at least potentially store the same amount of water and start with a similar storage amount, we define the starting point of integration of $dS^*$ using seasonal criteria. Therefore, we first plot $dS^*$ against accumulated annual precipitation normalized by the long-term mean annual precipitation $(MAP)$ (Fig. 10). This helps to compare states with similar potential accumulated input. Next, we re-scale the ordinate such that the origin corresponds to the mean of the local periodic minima, assuming that the soil moisture is near the permanent wilting point at these times. This way we gain a dimensionless estimator for the total active bulk catchment water storage. Values in $dS^*$ of around zero indicate dry conditions whereas values around 1 indicate that dynamic storage is equal to the root zone storage volume. Note that both values > 1 (e.g. during the occurrence of snow) and values < 0, may occur and that absolute values must not be interpreted. Please note, that we encountered significant trends and erratic fluctuations in $dS^*$ in the majority of all sites.

– FIGURE 10: coherent normalization of integrative storage measures –

### A2  Automated delineation of rainfall driven events

Comparability of runoff coefficients requires essentially an automated detection of rainfall-runoff events in continuous time series to pool enough events into a statistically analyzable sample. The concept and interpretation of runoff coefficients $(CR)$ on both event and annual time scales is old and dates back to Sherman (1932). Up to now $CRs$ are frequently used as diagnostic variables to describe





response properties and runoff generation (compare e.g., Pearce et al., 1986; Merz et al., 2006; Merz and Blöschl, 2009; Graeff et al., 2012; Capell et al., 2012, and many others). However, $CRs$ are not defined consistently (e.g. total runoff over total precipitation vs. total quick flow over total precipitation) and the literature describes a range of different methods for the detection of the start
and end of an event which are required for the separation of the slow flow component as illustrated by Blume et al. (2007). We extensively tested various approaches including baseflow separation and filtering techniques (e.g., Douglas and Peucker, 1973; Chapman, 1999; Perng et al., 2000; Eckhardt, 2005), penalty functions (Drabek, 2010), fuzzy logic (Seibert and Ehret, 2012), and the methods proposed by Merz and Blöschl (2009) and Norbiato et al. (2009). However, the results of these
methods were usually unsatisfactory when applied to a range of different regimes of precipitation and stream flow. In the end we adapt and recombine different existing techniques and detect rainfall-runoff events based upon the following principles: First, we select *rainfall events* as subsequent periods of liquid rainfall (maximum up to 6 h of rain free period are tolerated) with at least 10 mm of daily rain depth (compare also Fig. 11). Given these periods, we identify the corresponding *discharge*
*events* starting with the maximum flow rate. Between the latter and the beginning of rainfall we search for the first point in time where $dQ/dt > 0$ holds true for five subsequent time steps which we define as the start of the discharge event. Starting from the peak flow we next define the end of the discharge event using the constant-k method proposed by Blume et al. (2007). Due to missing convergence of this approach in 20 - 40 % of all cases we combine it with additional cut-off criteria
(e.g. threshold exceedance, beginning of next rainfall event, and others). Missing convergence often results from varying rainfall intensities throughout the event. In our data set we observed that the occurrence of multiple peaks and troughs within a "single" event is more often the rule rather than the exception. Upon request, a program code for the automated detection of rainfall-runoff events in hourly time series which is written in R (R Development Core Team, 2015) can be obtained from
the author.

– FIGURE 11: Automated detection of rainfall-runoff events –

**A3   Linkage between site identifiers and gauge names**

Table 6 relates the site identifiers we introduce in section 3 to the corresponding gauge and stream names.

– TABLE 6: Link table –

*Acknowledgements.*  We gratefully acknowledge data provision by the Bavarian Environmental Agency (LfU) and the German (DWD) and Austrian weather services (ZAMG). Funding was partly provided by the LfU which is gratefully acknowledged as well. We also thank Hoshin Gupta for the inspiring discussion on the





double and triple mass curve concepts a few years ago. We furthermore acknowledge support by Deutsche

Forschungsgemeinschaft and Open Access Publishing Fund of Karlsruhe Institute of Technology.





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



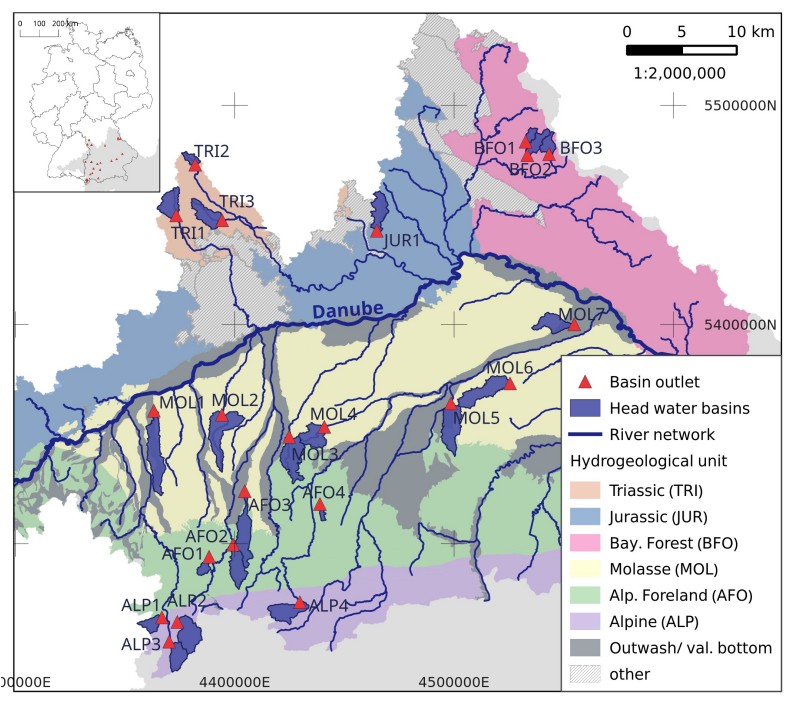

**Figure 1.** Upper Danube catchment in southern Germany with selected head water basins (blue polygons), corresponding gauges (red triangles) and major river network (blue lines). The site identifiers (IDs) refer to the corresponding (hydro)geological unit (color coded map in the background, adapted from BGR and SGD (2015)) and a single Arabic numeral. Moving from the North-West to the South we differentiate TRI (Triassic), JUR (Jurasic), BFO (Bavarian Forest), MOL (Faulted Molasse), AFO (Alpine Foreland) and ALP (Alpine). Appendix A3 provides links between the site IDs and the real gauge names. The inset in the upper left corner shows Germany's federal state boundaries, the individual head water outlets and the basin of the Danube (grey area). The grid coordinates refer to the Gauss-Kruger zone 4 projection (CRS identifier EPSG:31468).





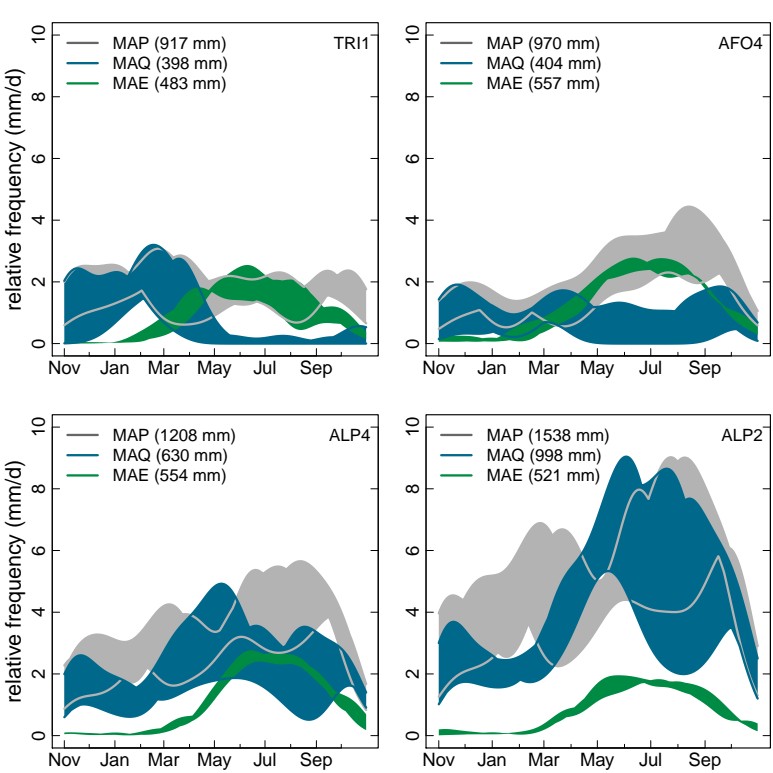

**Figure 2.** Kernel density estimates (regime curves) of observed areal precipitation (grey), discharge (blue) and calculated areal mean evapotranspiration (Penman-Monteith) (green) of four selected head water catchments. The width of the individual bands illustrates the inter-annual variation during the four year lasting period. In all cases identical kernels and bandwidths are used for variables of the same type. MAP, MAQ and MAE provide information on the four year mean annual average for P, Q and E respectively.





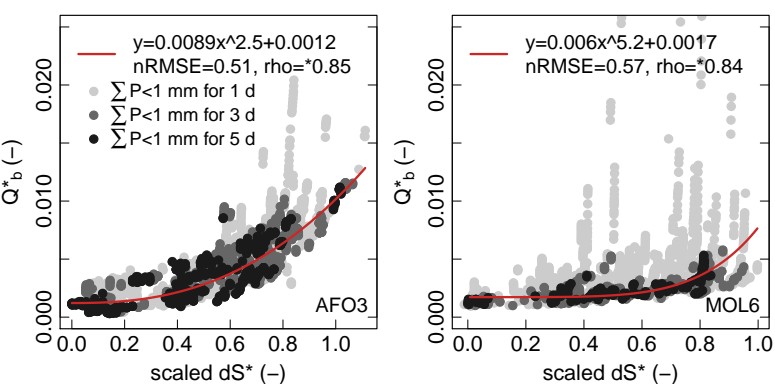

**Figure 3.** Normalized empirical storage-baseflow relationship for the catchments AFO3 (left) and MOL6 (right). $dS^*$ and $Q_b^*$ are calculated according to Eq. (1 and 4), respectively. The power law functions (red lines) are fitted to stream flow values where the last input in precipitation > 0.1 mm was $\geq$ 5 d ago (statistics in the top). The quality of each relation is judged using the root-mean-square-error normalized to the standard deviation of the sample (nRMSE) and Spearman's rank coefficient of correlation (rho). Note: Fitting the model to the data required a re-scaling of $dS^*$ to the range [0..∞] to prevent root extraction from negative values.




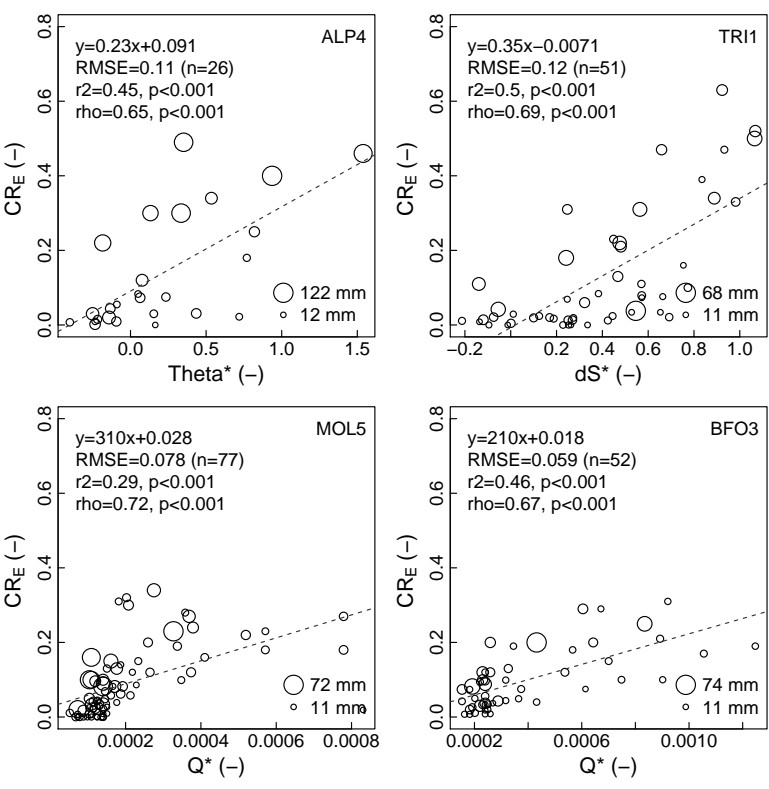

**Figure 4.** Event runoff coefficients ($CR_E$) plotted against the normalized storage measures $\theta^*$, $dS^*$ or $Q^*$ for the sites ALP4, TRI1, MOL5 and BFO3. The point sizes are scaled according to the corresponding rain depth. Statistical information is provided in terms of a regression (dotted line), its equation, the sample size (n), the root-mean-squared-error (RMSE), Pearsons' coefficient of determination ($r^2$), Spearmans' rank coefficient of correlation (rho) and corresponding p-values (p).





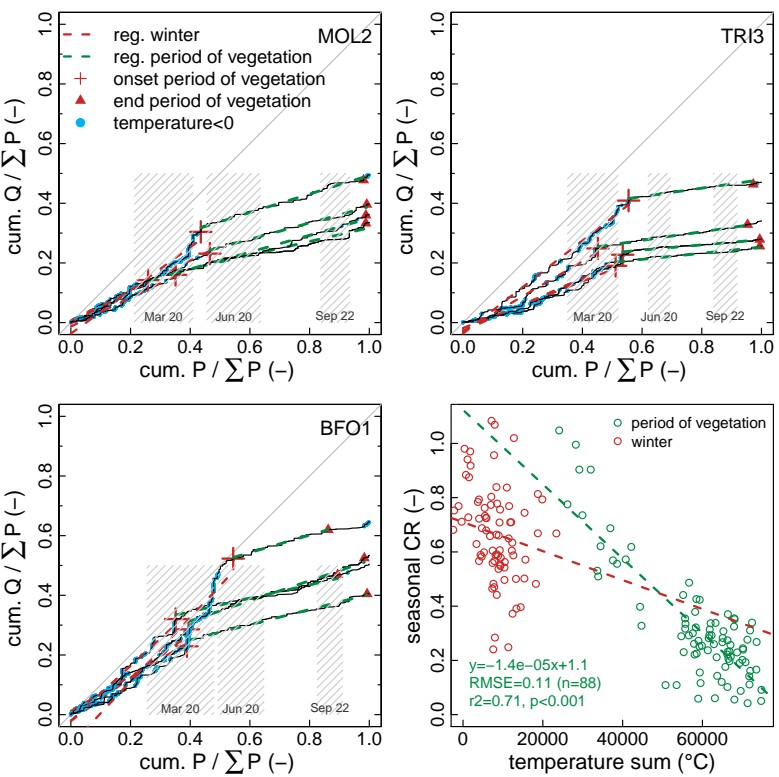

**Figure 5.** Normalized double mass curves (nDMC) for the catchments MOL2 (*top left*), TRI3 (*top right*) and BFO1 (*bottom left*) for the hydrological years 1999-2003. Onset and end of the period of vegetation are determined using a temperature index model. Regression lines are fitted to both periods (dotted lines in red/green), their slopes are interpreted as seasonal runoff coefficients. Periods with temperatures < 0 °C are highlight in blue. Gregorian definitions for the start of spring (Mar 20th), start of summer (Jun 20th) and start of fall (Sep 22th) (hatched polygons) are added to the $cum. P/\sum P$ plane to highlight their differences to temperature based estimates on the onset and end of the period of vegetation. Statistical properties of all nDMCs are summarized in Table 5. The *bottom right panel* shows seasonal hourly temperature sums (calculated for each hydrological year starting from Nov 1st) and corresponding seasonal runoff coefficients for all sites (n=22) and years (n=4). The dotted lines are regressions. Statistical information on the summer model is plotted in green. During winter there was no significant statistical relation available ($r^2$=0.04, p=0.062).





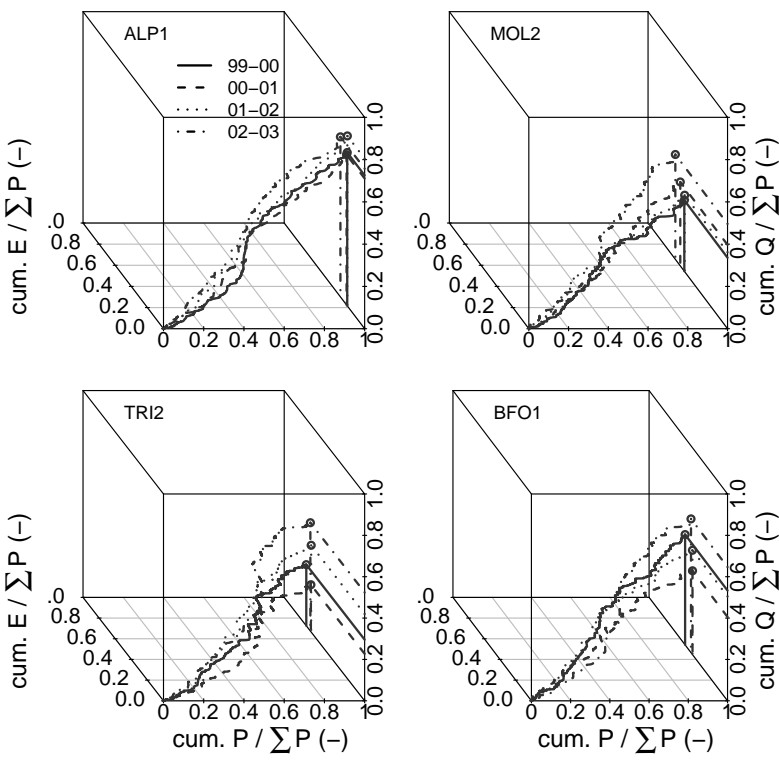

**Figure 6.** Normalized triple mass curves (nTMCs) from the catchments ALP1, MOL2, TRI2 and BFO1. Each plot contains data from four different hydrological years ('99-'00, '00-'01, '01-'02 and '02-'03) which are coded using different line styles. In the *upper row* (sites ALP1 and MOL2) the inter-annual variations in $cum. Q/\sum P$ are rather identical to the inter-annual variations in $cum. E/\sum P$. At TRI2 and BFO1 (*lower row*), the inter-annual variations in $cum. Q/\sum P$ are much larger than the inter-annual variations in $cum. E/\sum P$. Corresponding statistics of the normalized double and triple mass curves are summarized in Table 5.




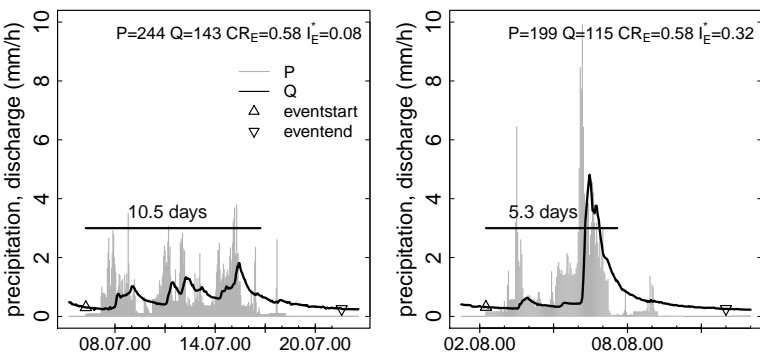

**Figure 7.** Two storm events from the alpine catchment ALP2 with almost similar total amounts of precipitation
(P) and discharge (Q), identical runoff coefficient ($CR_E$), but with different duration and thus, intensities.
The latter is reflected in different $I_E^*$ (Eq. 7) which corresponds to the normalized maximum temporal change
in intensity. In the first case (*left panel*) we expect that capacity controlled processes to dominate the runoff
generation. In the second case (*right panel*) we assume that the steep rising limb and the high peak discharge
are caused by intensity controlled runoff formation processes.





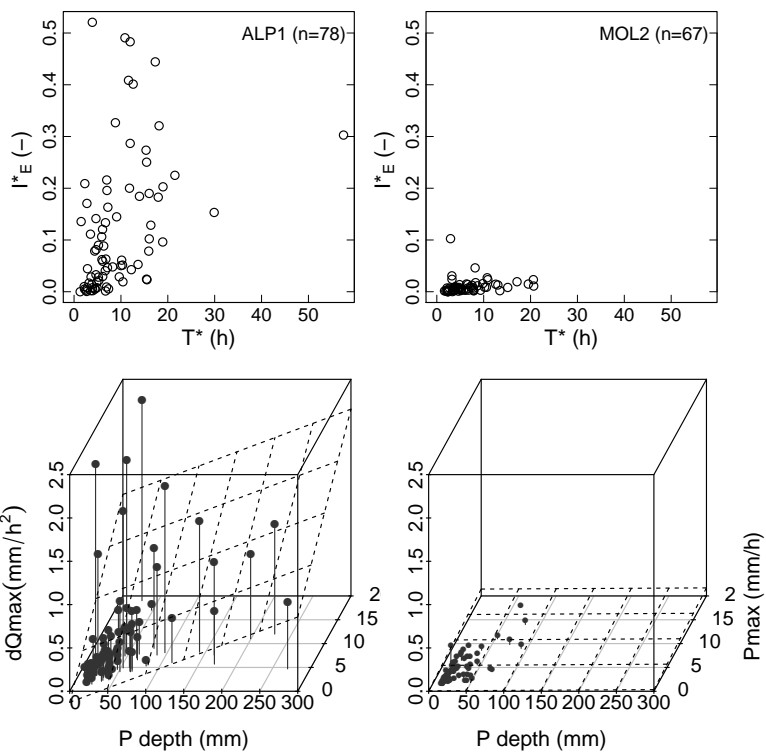

**Figure 8.** Diagnostics for the detection of intensity controlled conditions. The panels in the *upper row* show scatterplots of the normalized event duration $T_E^*$ (Eq. 6) which is plotted against the normalized temporal intensity changes $(I_E^*)$ (Eq. 7). The *lower row* shows corresponding three-dimensional scatterplots with event rain depth, maximum observed rain intensity and the maximum of the temporal derivative of observed discharge $(dI_{Q,max})$ are plotted on the x, y and z-axis respectively. The inclined plane (dotted) represents the plane of a multiple linear regression. The left column shows a conclusive case (site ALP1) where the frequent occurrence of intensity controlled runoff formation processes is likely. The right column shows inconclusive results from the site MOL2.





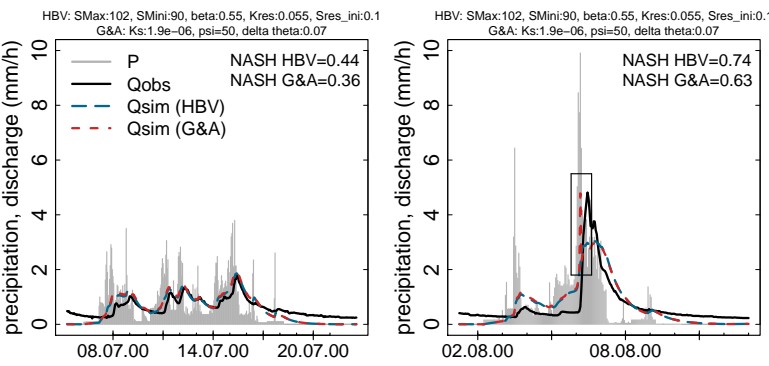

**Figure 9.** Simulated storm hydrographs from a capacity controlled event (*left*) and a (partly) intensity controlled event (*right*). Simulations use a HBV type betastore (red line) and a Green and Ampt (G&A) approach (blue line). Both concepts are combined with a linear reservoir. For the HBV type we optimize maximum storage depth ($SMax$), beta parameter ($beta$) and reservoir constant ($kres$) in an event based mode using a simulated annealing algorithm (initial fillings of the betastore $SMini$ and of the linear reservoir $Sres, ini$ are insensitive to the peak flow simulation accuracy). G&A is parametrized based upon a Rosetta (Schaap et al., 2001) estimate of $Ks$ and a literature value for the suction head ($psi$) at the wetting front (Maidment, 1993). The parameters of the linear reservoir ($SMax$ and $kres$) are adopted from the HBV optimization. As (statistical) reference for the model performance we provide the Nash-Sutcliffe-Efficiency (NASH) criterion. Statistics of the rainfall-runoff events are provided in Fig. 7.





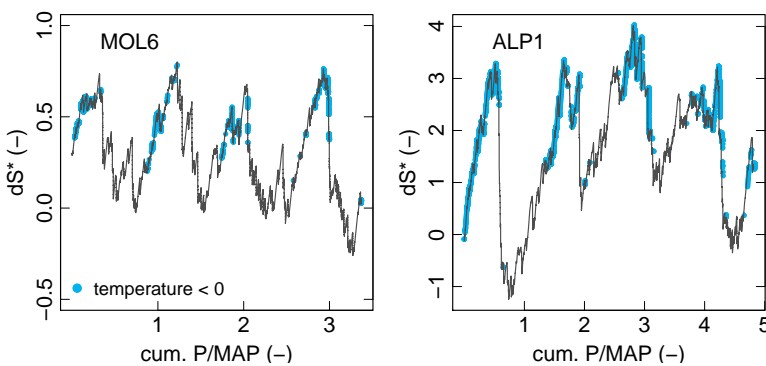

**Figure 10.** Coherent normalization of integrative storage measures. The plots show examples from the sites MOL6 and ALP1 for the same four year period (11/1999-10/2003). Re-scaled and normalized dynamic storage $dS^*$ (Eq. 1) is plotted on the ordinate, the abscissa shows cumulated precipitation divided by the long term mean annual precipitation ($MAP$). Please note the differences between the two sites in the scaling of the axes.





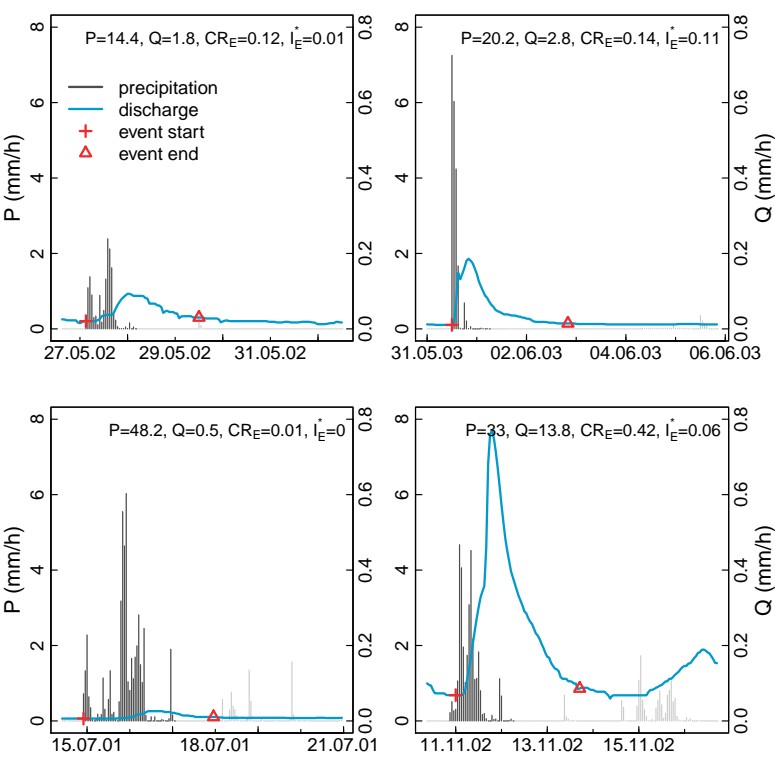

**Figure 11.** Automated detection of rainfall-runoff events. The plots show results from four selected events from the site TRI3. The examples are selected from different seasonal periods and illustrate different temporal dynamics of forcing and response. The statistics in the top provide event specific totals of rainfall $(P)$ and quickflow $(Q)$, the event runoff coefficient $CR_E$ (Eq. 5) and the normalized temporal intensity change $I_E^*$ (Eq. 7).





**Table 1.** Physiographic catchment properties in terms of topography, landuse and hydro-meteorology. The columns contain site identifier (ID), catchment size $A$, mean catchment elevation above sea level (elev), median gradient ($\phi$), median gradient times average saturated hyd. conductivity ($\phi \cdot Ks$), relative land coverage ratios for infrastructure (infr), arable land (arab), pasture (past), forest (frst), wetlands (wet) and rock outcrops (rock), the 30 year mean annual precipitation (MAP), four year mean annual precipitation ($\overline{P}$), discharge ($\overline{Q}$), runoff coefficient ($\overline{CR}$), streamflow coefficient of variation (var.c(Q)) and the slope of the flow duration curve between the 33 and 66% percentiles (sFDC). $\overline{CR}$, var.c(Q) and sFDC are dimensionless.

| ID | topography | | | | % land use coverage | | | | | | hydro-meteorology | | | | | |
|---|---|---|---|---|---|---|---|---|---|---|---|---|---|---|---|---|
| | $A$ [km²] | elev [m] | $\phi$ [-] | $\phi \cdot Ks$ [m/s] | infr | arab | past | frst | wet | rock | MAP [mm] | $\overline{P}$ [mm] | $\overline{Q}$ [mm] | $\overline{CR}$ | var.c(Q) | sFDC |
| TRI1 | 88 | 481 | 2.8e-02 | 1.0e-06 | 0.04 | 0.54 | 0.20 | 0.21 | 0 | 0 | 802 | 919 | 0.045 | 0.43 | 2.00 | -1.46 |
| TRI2 | 26 | 460 | 2.5e-02 | 9.8e-07 | 0 | 0.60 | 0.12 | 0.29 | 0 | 0 | 707 | 801 | 0.033 | 0.36 | 2.20 | -1.94 |
| TRI3 | 93 | 468 | 3.8e-02 | 8.3e-07 | 0.02 | 0.62 | 0.07 | 0.30 | 0 | 0 | 738 | 829 | 0.032 | 0.33 | 1.80 | -0.99 |
| JUR1 | 90 | 518 | 7.2e-02 | 2.0e-06 | 0.01 | 0.59 | 0.15 | 0.26 | 0 | 0 | 833 | 839 | 0.046 | 0.48 | 0.83 | -0.78 |
| BFO1 | 25 | 620 | 6.9e-02 | 2.5e-06 | 0 | 0.35 | 0.06 | 0.60 | 0 | 0 | 889 | 933 | 0.054 | 0.51 | 0.89 | -0.61 |
| BFO2 | 64 | 635 | 6.1e-02 | 1.3e-06 | 0.02 | 0.39 | 0.12 | 0.47 | 0.01 | 0 | 893 | 920 | 0.055 | 0.53 | 0.91 | -0.64 |
| BFO3 | 58 | 624 | 7.9e-02 | 1.2e-06 | 0.01 | 0.24 | 0.21 | 0.55 | 0 | 0 | 908 | 825 | 0.052 | 0.56 | 1.10 | -0.76 |
| MOL1 | 166 | 543 | 1.1e-02 | 1.6e-08 | 0.07 | 0.42 | 0.29 | 0.23 | 0 | 0 | 889 | 973 | 0.042 | 0.38 | 0.83 | -0.63 |
| MOL2 | 163 | 515 | 4.0e-02 | 6.9e-08 | 0.03 | 0.28 | 0.37 | 0.32 | 0 | 0 | 901 | 1010 | 0.045 | 0.39 | 0.90 | -0.36 |
| MOL3 | 163 | 558 | 1.4e-02 | 2.0e-08 | 0.05 | 0.69 | 0.10 | 0.15 | 0 | 0 | 933 | 1100 | 0.057 | 0.45 | 0.64 | -0.27 |
| MOL4 | 97 | 517 | 2.5e-02 | 3.9e-08 | 0.04 | 0.77 | 0.03 | 0.15 | 0 | 0 | 888 | 1016 | 0.042 | 0.36 | 0.93 | -0.50 |
| MOL5 | 133 | 473 | 2.0e-02 | 4.4e-08 | 0.05 | 0.81 | 0.05 | 0.09 | 0 | 0 | 883 | 1016 | 0.047 | 0.40 | 1.00 | -0.57 |
| MOL6 | 146 | 484 | 4.2e-02 | 7.8e-08 | 0.02 | 0.79 | 0.04 | 0.15 | 0 | 0 | 856 | 721 | 0.029 | 0.35 | 1.60 | -0.58 |
| MOL7 | 87 | 379 | 2.4e-02 | 3.4e-08 | 0.02 | 0.73 | 0.01 | 0.24 | 0 | 0 | 744 | 733 | 0.026 | 0.31 | 1.10 | -0.71 |
| AFO1 | 45 | 840 | 3.4e-02 | 3.6e-08 | 0.01 | 0.11 | 0.27 | 0.62 | 0 | 0 | 1388 | 1243 | 0.073 | 0.51 | 1.90 | -1.32 |
| AFO2 | 95 | 777 | 4.0e-02 | 9.1e-08 | 0.01 | 0.11 | 0.55 | 0.31 | 0.01 | 0 | 1292 | 1466 | 0.083 | 0.50 | 1.30 | -0.76 |
| AFO3 | 136 | 751 | 2.2e-02 | 5.2e-08 | 0.05 | 0.12 | 0.62 | 0.22 | 0 | 0 | 1198 | 1015 | 0.045 | 0.38 | 0.72 | -0.75 |
| AFO4 | 12 | 688 | 2.9e-02 | 3.2e-08 | 0.07 | 0.32 | 0.31 | 0.29 | 0 | 0 | 1114 | 1024 | 0.047 | 0.40 | 1.20 | -1.03 |
| ALP1 | 47 | 1279 | 3.3e-01 | 1.8e-06 | 0.00 | 0.05 | 0.50 | 0.45 | 0 | 0 | 2212 | 2662 | 0.230 | 0.75 | 1.40 | -1.14 |
| ALP2 | 127 | 1433 | 4.0e-01 | 7.7e-07 | 0.01 | 0.02 | 0.55 | 0.28 | 0 | 0.15 | 2315 | 2526 | 0.240 | 0.83 | 1.10 | -0.83 |
| ALP3 | 76 | 1539 | 5.1e-01 | 7.6e-07 | 0.01 | 0.01 | 0.59 | 0.18 | 0 | 0.21 | 2438 | 2181 | 0.210 | 0.86 | 0.89 | -1.07 |
| ALP4 | 114 | 1270 | 4.2e-01 | 5.1e-07 | 0.01 | 0.01 | 0.25 | 0.61 | 0.02 | 0.09 | 1826 | 1684 | 0.120 | 0.64 | 1.00 | -0.49 |





**Table 2.** Soil properties of the selected headwater catchments: Root zone depth ($\tau$), effective field capacity ($eFC_\tau$), air capacity ($AC_\tau$), field capacity ($FC_\tau$), total pore volume ($TPV_\tau$), and contents of clay, silt, sand and skeleton (skel). $FC$, $eFC$, $AC$ and $TPV$ refer to $\tau$. $nsoils$ gives information on the total number of different soils classes within the individual sites. Average saturated hydraulic conductivity ($Ks$) was estimated based on grain size data Schaap et al. (2001). The soil properties are weighted means (areal share) the national soil map of Germany (BGR, 1995). We also provide aquifer productivity classes ($APR$) from the international hydrogeological map of Europe (Duscher et al., 2015). The categorical $APR$ values 1, 2, 3 and 4 indicate dominance of highly productive, low to moderately productive conditions, dominance of locally aquiferous rocks and non-aquiferous rocks.

| SITE | $\tau$ [cm] | $eFC_\tau$ [mm] | $AC_\tau$ [mm] | $FC_\tau$ [mm] | $TPV_\tau$ [mm] | clay [%] | silt [%] | sand [%] | skel [%] | $Ks$ [m/s] | n soils | $APR$ |
|---|---|---|---|---|---|---|---|---|---|---|---|---|
| TRI1 | 68.3 | 73.9 | 47.1 | 262.3 | 309.4 | 42.7 | 9.3 | 48.6 | 1.8 | 3.6e-05 | 3 | 2 |
| TRI2 | 67.1 | 71.0 | 49.8 | 248.4 | 298.2 | 39.9 | 9.0 | 51.7 | 1.9 | 3.9e-05 | 3 | 1 |
| TRI3 | 76.1 | 94.2 | 59.2 | 216.9 | 276.2 | 23.3 | 18.0 | 58.4 | 2.5 | 2.2e-05 | 4 | 1/2 |
| JUR1 | 26.2 | 32.5 | 37.0 | 71.0 | 108.0 | 39.2 | 16.8 | 43.5 | 3.7 | 2.8e-05 | 6 | 1 |
| BFO1 | 60.0 | 74.0 | 55.5 | 110.4 | 165.9 | 7.5 | 9.5 | 83.2 | 4.0 | 3.6e-05 | 5 | 4 |
| BFO2 | 54.8 | 74.5 | 40.8 | 124.0 | 164.8 | 12.4 | 17.0 | 71.1 | 3.6 | 2.1e-05 | 6 | 4 |
| BFO3 | 58.0 | 67.8 | 36.4 | 125.3 | 161.7 | 14.3 | 18.7 | 67.6 | 3.8 | 1.5e-05 | 5 | 4 |
| MOL1 | 85.1 | 144.8 | 44.6 | 330.9 | 375.5 | 26.4 | 58.8 | 14.4 | 1.1 | 1.5e-06 | 8 | 1/2/3 |
| MOL2 | 78.9 | 146.1 | 44.3 | 312.1 | 356.4 | 22.3 | 56.5 | 21.8 | 1.4 | 1.7e-06 | 4 | 2 |
| MOL3 | 86.5 | 151.3 | 55.4 | 290.7 | 346.1 | 22.0 | 48.5 | 28.9 | 1.9 | 1.5e-06 | 8 | 2 |
| MOL4 | 89.2 | 132.5 | 46.2 | 323.9 | 370.2 | 22.7 | 57.2 | 21.0 | 1.2 | 1.6e-06 | 4 | 2 |
| MOL5 | 88.2 | 138.3 | 54.3 | 288.8 | 343.0 | 19.0 | 46.5 | 34.8 | 1.9 | 2.2e-06 | 12 | 1/2 |
| MOL6 | 88.3 | 129.5 | 47.7 | 308.3 | 355.9 | 21.0 | 54.0 | 24.7 | 1.3 | 1.9e-06 | 3 | 2 |
| MOL7 | 90.0 | 167.1 | 47.5 | 328.6 | 376.1 | 23.0 | 66.8 | 11.2 | 1.0 | 1.4e-06 | 3 | 2/3 |
| AFO1 | 100 | 153.0 | 76.0 | 322.5 | 398.5 | 21.0 | 39.0 | 40.0 | 2.0 | 1.1e-06 | 1 | 3 |
| AFO2 | 74.0 | 149.6 | 57.0 | 285.8 | 342.8 | 19.8 | 39.5 | 40.8 | 2.2 | 2.3e-06 | 5 | 3 |
| AFO3 | 79.4 | 138.4 | 61.8 | 266.0 | 327.8 | 19.3 | 33.7 | 47.1 | 2.5 | 2.4e-06 | 8 | 3 |
| AFO4 | 80.0 | 142.2 | 59.5 | 283.4 | 342.9 | 20.8 | 38.0 | 41.2 | 1.8 | 1.1e-06 | 2 | 2 |
| ALP1 | 55.5 | 90.2 | 38.5 | 209.3 | 247.8 | 15.8 | 23.7 | 60.5 | 2.5 | 5.5e-06 | 2 | 2/3 |
| ALP2 | 28.1 | 42.2 | 18.8 | 108.0 | 126.7 | 28.2 | 28.8 | 44.5 | 4.2 | 1.9e-06 | 5 | 1/2 |
| ALP3 | 24.1 | 35.2 | 16.0 | 92.7 | 108.7 | 29.4 | 28.7 | 42.4 | 4.4 | 1.5e-06 | 5 | 1/2 |
| ALP4 | 28.9 | 41.1 | 18.7 | 110.7 | 129.4 | 37.5 | 24.4 | 39.3 | 4.3 | 1.2e-06 | 5 | 1 |





**Table 3.** Statistics of the radiation-driven case (baseflow) where the last input in precipitation > 1 mm is at least five days ago: The table contains the sample size (n), Spearmans rank coefficients of correlation ($\rho$) between the storage measures $dS^*$ and $\theta^*$, between storage measures and normalized specific stream flow depths ($Q_b^*$) and multiplier and exponent of the fitted non-linear model. As a quality of fit criterion for the latter we provide the root-mean-squared-error normalized by the standard deviation of the sample (nRMSE).

| Site | n | $\rho$ between | | | non-linear model | | |
|---|---|---|---|---|---|---|---|
| | | $dS^* \& \theta^*$ | $Q_b^* \& dS^*$ | $Q_b^* \& \theta^*$ | multiplier | exponent | nRMSE |
| TRI1 | 364 | *0.2 | *0.67 | 0.01 | 1.8e-04 | 1.3 | 0.56 |
| TRI2 | 384 | -0.01 | *0.41 | *0.18 | 1.8e-07 | 7.2 | 0.89 |
| TRI3 | 397 | *-0.27 | *0.88 | -0.02 | 1.4e-04 | 1.8 | 0.44 |
| JUR1 | 588 | 0.13 | *0.75 | -0.01 | 4.4e-05 | 2.5 | 0.70 |
| BFO1 | 1265 | 0.09 | *0.56 | 0.06 | 7.9e-05 | 2.3 | 0.65 |
| BFO2 | 1235 | 0.02 | *0.69 | 0.03 | 1.5e-04 | 1.6 | 0.68 |
| BFO3 | 899 | *0.30 | *0.47 | -0.01 | 3.7e-04 | 0.8 | 0.87 |
| MOL1 | 968 | *0.39 | *0.84 | *0.16 | 8.1e-03 | 1.1 | 0.61 |
| MOL2 | 654 | *-0.24 | *0.49 | 0.10 | 8.8e-04 | 2.0 | 0.84 |
| MOL3 | 447 | -0.02 | *0.86 | -0.11 | 5.4e-03 | 1.6 | 0.59 |
| MOL4 | 634 | -0.11 | *0.50 | *0.26 | 1.7e-03 | 0.9 | 0.84 |
| MOL5 | 1184 | *0.18 | *0.67 | *0.22 | 2.9e-03 | 0.7 | 0.83 |
| MOL6 | 213 | *0.30 | *0.84 | *0.28 | 6.0e-03 | 5.2 | 0.57 |
| MOL7 | 1018 | *-0.28 | *0.16 | -0.07 | 1.1e-03 | 0.8 | 1.01 |
| AFO1 | 250 | 0.16 | *0.45 | 0.14 | 5.0e-03 | 0.3 | 0.95 |
| AFO2 | 609 | 0.04 | 0.12 | *0.54 | 7.8e-04 | 15.0 | 0.92 |
| AFO3 | 708 | *0.21 | *0.85 | 0.07 | 8.9e-03 | 2.5 | 0.49 |
| AFO4 | 356 | 0.05 | *0.42 | *0.22 | 8.8e-03 | 0.6 | 0.86 |
| ALP1 | 90 | NA | NA | NA | NA | NA | NA |
| ALP2 | 116 | 0.03 | *0.66 | 0.19 | 5.5e-04 | 2.2 | 0.64 |
| ALP3 | 249 | 0.15 | *0.60 | 0.10 | 1.0e-08 | 6.6 | 0.43 |
| ALP4 | 80 | NA | NA | NA | NA | NA | NA |
| * cases | - | 9 | 19 | 7 | - | - | - |

*-symbols code significant values (p-values<0.001). Note: Snow and frequent rainfall yielded small (n<100) and highly skewed samples in two alpine basins (ALP1 and ALP4). There values are set to NA as a meaningful interpretation is not feasible.



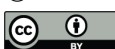

**Table 4.** Statistics of rainfall-driven conditions: Spearman rank coefficients of correlation ($\rho$) and Pearsons coefficient of determination ($r^2$) between the event runoff coefficients ($CR_E$) and the three different storage measures ($dS^*$, $\theta^*$ and $Q^*$), between the storage measures themselves and, results for a multiple linear regression (equation and corresponding $r^2$) between $CR_E$ and the two most explanatory (uncorrelated) storage measures. If all three storage measures were correlated significantly, both, the equation and the $r^2$ value were set to NA. We also provide the slope of the linear regression ($b$) between $CR_E$ and $Q^*$ for cases were the latter were correlated significantly.

| Site | n | $CR_E$ & $dS^*$ $\rho$ | $r^2$ | $CR_E$ & $Q^*$ $\rho$ | $r^2$ | $CR_E$ & $\theta^*$ $\rho$ | $r^2$ | $dS^*$ & $\theta^*$ $\rho$ | $r^2$ | $Q^*$ & $dS^*$ $\rho$ | $r^2$ | $Q^*$ & $\theta^*$ $\rho$ | $r^2$ | lin reg. $b$ | multiple lin. regression equation | $r^2$ |
|---|---|---|---|---|---|---|---|---|---|---|---|---|---|---|---|---|
| TRI1 | 51 | *0.69 | *0.50 | *0.81 | *0.73 | 0.38 | 0.18 | 0.35 | 0.11 | *0.78 | *0.49 | 0.26 | 0.18 | 2.94 | *340$Q^*$+0.087$\theta^*$ | 0.73 |
| TRI2 | 40 | *0.57 | *0.27 | *0.82 | *0.57 | 0.44 | 0.14 | 0.40 | 0.11 | *0.53 | 0.24 | 0.44 | *0.30 | 3.87 | *480$Q^*$-0.086$\theta^*$ | 0.57 |
| TRI3 | 37 | *0.77 | *0.46 | *0.88 | *0.59 | 0.39 | 0.13 | 0.15 | 0.01 | *0.83 | *0.36 | 0.45 | 0.26 | 3.13 | *500$Q^*$-0.069$\theta^*$ | 0.59 |
| JUR1 | 40 | 0.39 | 0.20 | *0.63 | *0.29 | 0.44 | 0.08 | 0.41 | 0.20 | *0.63 | *0.46 | *0.52 | *0.43 | 1.29 | NA | NA |
| BFO1 | 67 | 0.30 | 0.17 | *0.46 | *0.16 | 0.35 | 0.07 | 0.37 | 0.14 | 0.38 | 0.27 | *0.54 | *0.27 | 1.09 | 91$Q^*$+0.034$dS^*$ | 0.22 |
| BFO2 | 65 | *0.51 | *0.31 | *0.49 | *0.24 | *0.41 | 0.09 | 0.38 | 0.14 | *0.73 | *0.46 | *0.41 | *0.26 | 1.22 | *0.061$dS^*$+0.057$\theta^*$ | 0.32 |
| BFO3 | 52 | *0.71 | *0.47 | *0.67 | *0.46 | *0.55 | *0.25 | 0.43 | 0.19 | *0.60 | *0.43 | *0.51 | *0.33 | 1.97 | *0.085$dS^*$+0.14$\theta^*$ | 0.52 |
| MOL1 | 66 | *0.62 | *0.23 | *0.67 | *0.35 | 0.32 | 0.05 | 0.23 | 0.04 | *0.85 | *0.63 | 0.29 | 0.03 | 2.68 | *490$Q^*$+0.13$\theta^*$ | 0.37 |
| MOL2 | 67 | 0.26 | 0.07 | *0.52 | *0.19 | 0.18 | 0.03 | 0.11 | 0 | *0.52 | *0.24 | *0.47 | *0.24 | 1.86 | NA | NA |
| MOL3 | 73 | *0.49 | *0.22 | *0.49 | *0.20 | 0.34 | 0.06 | 0.19 | 0.01 | *0.73 | *0.46 | 0.38 | *0.17 | 1.47 | *0.097$dS^*$+0.14$\theta^*$ | 0.26 |
| MOL4 | 70 | 0.18 | 0.07 | *0.56 | *0.20 | *0.55 | *0.18 | 0.19 | 0.06 | *0.61 | *0.32 | *0.49 | *0.29 | 1.90 | NA | NA |
| MOL5 | 77 | *0.55 | *0.30 | *0.72 | *0.29 | *0.56 | *0.24 | *0.56 | *0.21 | *0.74 | *0.46 | *0.56 | *0.31 | 1.60 | NA | NA |
| MOL6 | 40 | 0.42 | 0.20 | 0.49 | 0.24 | 0.30 | 0.08 | 0.25 | 0.04 | *0.51 | 0.15 | *0.61 | *0.48 | NA | NA | NA |
| MOL7 | 38 | 0.30 | 0.16 | 0.48 | 0.26 | 0.50 | *0.28 | 0.15 | 0.01 | -0.08 | 0 | 0.36 | 0.22 | NA | 0.42$\theta^*$+300$Q^*$ | 0.37 |
| AFO1 | 54 | 0.01 | 0 | *0.64 | *0.27 | *0.57 | *0.26 | 0.23 | 0.04 | -0.05 | 0.01 | *0.58 | *0.53 | 1.65 | *380$Q^*$-0.059$dS^*$ | 0.28 |
| AFO2 | 61 | 0.19 | 0.04 | 0.48 | 0.07 | 0.33 | 0.07 | 0.29 | 0.03 | 0.25 | 0.06 | 0.39 | *0.24 | 0.75 | 100$Q^*$+0.26$\theta^*$ | 0.09 |
| AFO3 | 38 | 0.51 | *0.31 | 0.45 | 0.24 | 0.16 | 0.03 | 0.15 | 0 | *0.89 | *0.69 | 0.17 | 0.02 | NA | *0.11$dS^*$+0.085$\theta^*$ | 0.33 |
| AFO4 | 65 | *0.55 | *0.25 | *0.58 | *0.20 | *0.45 | 0.11 | 0.30 | 0.06 | *0.55 | *0.30 | *0.41 | *0.21 | 1.41 | NA | NA |
| ALP1 | 78 | 0.32 | 0.09 | 0.30 | 0.09 | *0.38 | 0.12 | 0.34 | *0.18 | 0.22 | 0.09 | *0.42 | *0.40 | NA | 0.14$\theta^*$+0.034$dS^*$ | 0.15 |
| ALP2 | 39 | 0.30 | 0.15 | 0.42 | 0.20 | 0.39 | 0.17 | 0.13 | 0.01 | 0.29 | 0.13 | *0.57 | *0.28 | NA | 31$Q^*$+0.04$dS^*$ | 0.26 |
| ALP3 | 30 | 0.52 | 0.22 | 0.01 | 0.06 | 0.41 | *0.34 | 0.30 | 0.13 | 0.24 | 0.14 | 0.41 | 0.19 | NA | 0.016$dS^*$+0.18$\theta^*$ | 0.42 |
| ALP4 | 26 | 0.13 | 0 | 0.23 | 0.15 | *0.72 | *0.50 | 0.33 | 0.12 | 0.07 | 0.03 | 0.29 | 0.28 | NA | *0.24$\theta^*$+1.8$Q^*$ | 0.5 |
| * count | - | 9 | 11 | 15 | 14 | 8 | 7 | 1 | 2 | 14 | 13 | 12 | 15 | - | - | - |

*-symbols code significant $\rho$ and $r^2$ values (p-value<0.001) and significant regressors in the case of the multiple lin. regression.





**Table 5.** Mean seasonal winter ($CR_W$), summer ($CR_S$) and annual runoff coefficients ($CR_{yr}$) as indicated by the slope of regression lines fitted to the normalized double mass curves. $CE_{yr}$ represents the mean annual evapotranspiration ratio. The inter-annual variations of these quantities within the hydrological years ('99-'03) is quantified using the mean absolute deviation which we provide by $mad_{CR_W}$, $mad_{CR_S}$, $mad_{CR_{YR}}$ and $mad_{CE_{YR}}$, respectively. All quantities are dimensionless.

| Site | $CR_W$ | $CR_S$ | $CR_{yr}$ | $CE_{yr}$ | $mad_{CR_W}$ | $mad_{CR_S}$ | $mad_{CR_{YR}}$ | $mad_{CE_W}$ |
|------|--------|--------|-----------|-----------|--------------|--------------|-----------------|--------------|
| TRI1 | 0.72 | 0.12 | 0.43 | 0.53 | 0.058 | 0.030 | 0.031 | 0.018 |
| TRI2 | 0.70 | 0.07 | 0.37 | 0.68 | 0.198 | 0.027 | 0.105 | 0.021 |
| TRI3 | 0.55 | 0.12 | 0.34 | 0.63 | 0.133 | 0.024 | 0.069 | 0.022 |
| JUR1 | 0.73 | 0.25 | 0.48 | 0.56 | 0.150 | 0.017 | 0.054 | 0.015 |
| BFO1 | 0.82 | 0.28 | 0.52 | 0.48 | 0.169 | 0.037 | 0.068 | 0.033 |
| BFO2 | 0.85 | 0.30 | 0.54 | 0.47 | 0.143 | 0.020 | 0.067 | 0.029 |
| BFO3 | 0.93 | 0.29 | 0.57 | 0.51 | 0.173 | 0.034 | 0.089 | 0.024 |
| MOL1 | 0.60 | 0.24 | 0.38 | 0.62 | 0.103 | 0.040 | 0.021 | 0.023 |
| MOL2 | 0.56 | 0.27 | 0.40 | 0.58 | 0.080 | 0.022 | 0.049 | 0.042 |
| MOL3 | 0.62 | 0.34 | 0.46 | 0.57 | 0.069 | 0.019 | 0.045 | 0.035 |
| MOL4 | 0.56 | 0.23 | 0.37 | 0.61 | 0.084 | 0.021 | 0.051 | 0.048 |
| MOL5 | 0.69 | 0.22 | 0.41 | 0.58 | 0.055 | 0.028 | 0.026 | 0.023 |
| MOL6 | 0.60 | 0.20 | 0.36 | 0.66 | 0.066 | 0.018 | 0.025 | 0.015 |
| MOL7 | 0.35 | 0.27 | 0.31 | 0.68 | 0.139 | 0.089 | 0.031 | 0.052 |
| AFO1 | 0.72 | 0.35 | 0.51 | 0.46 | 0.031 | 0.120 | 0.042 | 0.053 |
| AFO2 | 0.68 | 0.34 | 0.49 | 0.48 | 0.036 | 0.099 | 0.043 | 0.052 |
| AFO3 | 0.56 | 0.24 | 0.38 | 0.62 | 0.130 | 0.068 | 0.031 | 0.056 |
| AFO4 | 0.66 | 0.22 | 0.39 | 0.64 | 0.050 | 0.090 | 0.030 | 0.092 |
| ALP1 | 0.89 | 0.50 | 0.75 | 0.24 | 0.098 | 0.088 | 0.031 | 0.031 |
| ALP2 | 0.71 | 0.82 | 0.83 | 0.17 | 0.067 | 0.161 | 0.047 | 0.015 |
| ALP3 | 0.64 | 0.84 | 0.86 | 0.18 | 0.082 | 0.109 | 0.025 | 0.017 |
| ALP4 | 0.66 | 0.53 | 0.64 | 0.33 | 0.064 | 0.066 | 0.032 | 0.051 |
| mean | 0.67 | 0.32 | 0.49 | 0.51 | 0.10 | 0.06 | 0.046 | 0.035 |





**Table 6.** Link table that relates the site identifiers (ID) introduced in section 3 to the corresponding gauge and stream names. Gauge locations are provided in Gauß-Krüger zone 4 coordinates (GKR and GKH).

| ID | Gauge | Stream | GKR | GKH |
|------|-------|--------|-----|-----|
| TRI1 | Reichenbach (REIB) | Wörnitz | 4373327 | 5449863 |
| TRI2 | Binzwangen (BINZ) | Altmühl | 4381996 | 5473002 |
| TRI3 | Bechhofen (BECH) | Wieseth | 4394270 | 5447640 |
| JUR1 | Holnstein (HOLN) | Unterbürger Laber | 4464800 | 5442860 |
| BFO1 | Gartenried (GART) | Murach | 4532661 | 5483477 |
| BFO2 | Untereppenried (UEPR) | Ascha | 4533425 | 5477338 |
| BFO3 | Tiefenbach (TIEF) | Bayerische Schwarzach | 4543360 | 5477800 |
| MOL1 | Roth (ROTR) | Roth | 4363140 | 5360723 |
| MOL2 | Fleinhausen (FLEI) | Zusam | 4394141 | 5358887 |
| MOL3 | Mering (MERI) | Paar | 4424840 | 5348870 |
| MOL4 | Odelzhausen (ODZH) | Glonn | 4440860 | 5353360 |
| MOL5 | Appolding (APPO) | Strogen | 4498575 | 5364071 |
| MOL6 | Dietelskirchen (DIKI) | Kleine Vils | 4525540 | 5373175 |
| MOL7 | Wallersdorf (WALR) | Reißingerbach | 4554850 | 5400160 |
| AFO1 | Unterthingau (alt) (UTHI) | Kirnach | 4388313 | 5294058 |
| AFO2 | Hörmanshofen (HOER) | Geltnach | 4399272 | 5299593 |
| AFO3 | Buchloe (BUCH) | Gennach | 4404574 | 5323974 |
| AFO4 | Herrsching (HERR) | Kienbach | 4438860 | 5318140 |
| ALP1 | Gunzesried (GZRI) | Gunzesrieder Ach | 4366798 | 5266382 |
| ALP2 | Reckenberg (RECK) | Ostrach | 4373822 | 5264305 |
| ALP3 | Oberstdorf (OBTR) | Trettach | 4370128 | 5255320 |
| ALP4 | Oberammergau (OAMM) | Ammer | 4429723 | 5273332 |