# Peer review of "Unraveling abiotic and biotic controls on the seasonal water balance using data-driven dimensionless diagnostics"

_Hydrology and Earth System Sciences, 2016_

## Referee Comment (RC1) · Anonymous Referee #1 · 16 Mar 2016

Review of Seibert et al

Seibert et al, explore how well several dimensionless signatures can characterize the hydrologic response for 22 catchments in located in the Danube basin. The overarching goal of that analysis is to better understand the connections between state, structure and runoff behavior. Results indicate that certain signatures provide information about the hydrologic response from a catchment, suggesting functional similarity.

The (broad) goal of better understanding catchment similarity, using dimensionless signatures that represent our perception on underlying structural controls is in principle a relevant topic for HESS.

However, I do have several concerns that for me lead to the conclusion that the paper currently does not meet the standards of HESS. The overall lack a logically and smoothly flowing structure, and details are often unclear. This makes the paper difficult and time consuming to read,wherey the scientific novelties are difficult to distil.

Before publication in HESS the following items need to be addressed:

1) I cannot find a clear main novel contribution of this paper from either the abstract or the conclusion. This may be challenging to formulate as the paper tries to understand catchment similarity for different time-scales and processes, but I do expect that there are some overarching goals.

(i) The abstract is a mess: first you state a very broad and poorly defined problem, subsequently you state many details of the analysis that are not informative is the problem is still unclear. For example how is the reader supposed to know what you mean by "extensive/additive" and "intensive/non-additive". After you introduced the methods, your results focus on differences in functioning at different time-scales. Why didn't you introduce that you are looking at different timescales in your problem statement or methods? Your results do not clearly link to the introduction and methods part of your paper, and read like things you ad-hoc found and listed. You start stating results "Our dimensionless signatures evidently detect..." before the reader knows what data this comes from. This seems like an illogical and confusion order to me. Your conclusions feel like a random list of unconnected findings. Do you really need so many words for your abstract? I think you can (and should) be more to the point.

2) The introduction section lacks a logical build-up to a problem statement. Although I acknowledge that the introduction is giving a relevant overview of many challenges we face in concisely characterizing how catchments function, it reads like a long list of problems, rather than a structured introduction working defining a clear problem and working towards a clear goal.

3) Are you sure that "Question 2: Can we detect intensity controlled runoff formation

as essentially a high frequency process based on low frequency data?" is a clearly defined question for the (ignorant) reader (like me).?

4) What is the coherence of Q1-Q3 beyond "testing dimensionless measures to discriminate differences in runoff generation"? If there is coherence, please formulate it such that this is clear. If there is no coherence, why would you address these three questions within a single manuscript?

5) The section "Conceptual framework and candidate diagnostics" needs to be improved. Although you list requirements of the paper in the different subsection the writing is messy. I think you should be much more to the point in clearly stating the requirements. Problems that I came across and

a. "Requirements of functional diagnostics" is mainly an overview of your perception of how the hydrology of different landscapes function. Right now the reader reads one sentence about the "requirements" but are confused by the end of the section 2.1 what the purpose of this section is. b. The requirements of "Normalization of states and response measures" and ""Coherence and quality of integral storage measures" should be concisely written and to the point. c. I understand that you choose the metrics in section 2.2 based on all the requirements you have set before, but when you introduce them in this section I expect that the rationale of choosing them is clear. Currently it is not. For example "Lastly we use a normalized specific pre-event discharge (Q ) averaged across the last seven days:" is just an announcement, but leaves it completely unclear where the rationale to choose this metric comes from. The introduction of other metrics suffers from similar issues.

6) Is it logical that you provide the details of your study area only after you have discussed all the metric you use? I presume these metrics are largely based on your perception how the study catchment function. Therefor I think it would be more logical to switch their order.

7) There should be a story in your results, rather than that it is a long list of ad-hoc

results you have obtained. For example you begin with the statement that "During low flow conditions, the storage estimators dS and are in most cases linearly independent." And subsequently you list the associate statistics. The logic in your story is impossible to follow (for me). Listing the how the storage metrics link to flow characteristics is fine, but please make it more read like a story.

1) The paper presents the findings as "generally applicable for meso-scale catchments". This is an extremely bold statement since only 22 catchments in a very small part of the world are used.

8) The paper is uses many words, while I am not against long papers, it does feel that with 30% less words the story can be told too.

The current version of the manuscript is not clear enough to allow me to properly evaluate the results obtained in the paper. Therefor I suggest that this paper can only be considered for HESS after rigorous rewriting and restructuring of the paper has been done. I am sure that interesting new findings are available within the results, but at this stage I can not properly evaluate this.

Detailed comments

Introduction 1.1

(i) Line 35-36: remove one of the 2 "essentially" (ii) The statement that the current crude paradigm is "Hence, identical inputs of energy and rainfall will cause an identical runoff response, if two identical catchments are in the same state." seems just wrong to me. There is NO hydrologists that think all landscapes filter climate input the same way. I.e. structural setup matters is the paradigm. (iii) Why "particularly at the lower mesoscale."?

1.2.

(i) "Part of the confusion": which "confusion"? (ii) The section title suggests this is relevant at the meso-scale. I do not see the explanation why this is particular relevant

at this scale (compared to other catchments scales).

1.3

(i) Are you sure about the statement "and equally easily be upscaled".? I can think of a 1000 problems you can run into... (ii) Isn't "absolute storage" also difficult to work with as we can't observe it (at the catchment scale). (iii) If you introduce many terms that are not universally defined, such as "dynamic storage" and you don't explain what it means this can cause confusion/frustration for your reader. (iv) "it are the potential gradients" (not "is")

1.4

(i) To my knowledge Budyko, 1961 does NOT talk about the Budyko curve. A more appropriate reference would be his book "Climate and life" or the 1948 publication where he first mentions the curve (ii) "Although all these measures and their normalization can in principle be determined as residuals of the water balance and from available maps, the devil lies in the details as further elaborated in section 2.1." Can you structure your paper that we do not need such a "cliffhanger"? By the time your reader will be at section 2.1 he/she has forgotten about this statement.

1.5

(i) "mesoscale catchments". Since this scale is central in your paper, it would be good to give a definition of this scale (since to my knowledge a universal agreement on the definition is not common knowledge). (ii) I think the paper can benefit from more explicit linkages between the research questions.

2.1

(i) be sensitive to WHICH limiting factors? (ii) strong gradients in WHAT? I presume topographic? (iii) "Capacity controlled" or "Storage capacity controlled"?

2.1.1-2

(i) I had difficulty to efficiently read section 2.1.1 and 2.1.2

3.1

(i) are the scales of the maps (e.g. 1:1,000,0000) relevant, or is it the grid size that matters? (ii) Why are you confident that "LARSIM (Ludwig and Bremicker, 2006) . . . provides consistent areal estimates of evaporation, rainfall and snow water equivalent"?

3.2

(i) Lindsay, 2014; remove initials

4 4.1

(i) "when leaving out two alpine sites." What is your motivation/justification here? (ii) I read a long list of correlations you found, but can you make this a bit more into a story/take the reader by its hand?

4.2

(i) What do you mean by "of the driven case"

5.5.

(i) In your conclusions I expect that you answer question you have asked before. Currently it only lists recommendations of signatures to use without any context. (ii) "to discriminate differences in terrestrial runoff production in lower-mesoscale catchments". This might be very different for other regions with other climates and landscapes, so I do not think you can make this generalization. (iii) A reader will not (fully) understand your conclusions/outlook without reading the rest of the paper. That's not wrong, but probably not beneficial for you or the reader either.

Figures - Figure 2: if your y-axes states "relative frequency" I would not expect it unit's to be (mm/d). Or is this correct? I expect the a "Kernel density estimate" to be a non-parametric way to express the probability density function of the fluxes. - Figure 3:

replace "rho" by its symbol and "ˆ" by a upperscript - Figure 7&11: specify the units for the event totals

---

## Author Comment (AC1) · 23 Mar 2016

We thank the anonymous reviewer for the comments as he/she raised some important points that will help us to improve the manuscript. In the following we briefly reply to the main issues raised by the reviewer.

We regret that the structure of the manuscript hampered the reviewer to make his way to the innovative aspects of our study – we will stream line the revised manuscript and particularly the abstract as recommended by the reviewer.

We apologize for not having properly explained the terms "intensive" and "extensive" state variables, these are well known within environmental physics and thermodynamics (Zehe et al., 2014). Intensive state variables such as temperature, pressure of soil water potential are continuous at interfaces but not additive – if two rooms of the same temperature are connected by opening the door, temperature of the "merged system" stays the same, while the thermal energy is the sum of both energies. Intensive state variables in hydrology are matric potential, soil water velocities, rainfall intensities, while extensive state variables such as storage volumes, rainfall totals, etc. are extensive state variables and thus additive. A full characterization of a systems state implies that conjugated pairs of (extensive and intensive) state variables are known (Zehe et al., 2013) for instance the soil water content and the matric potential.

The relevance to our objective - the search for plots of normalized response measures against normalized state measures is that proper normalization depends on the nature of the runoff process. Storage controlled runoff processes are controlled/limited by storage which performs additive and can e.g. be estimated as residual of the water balance. Intensity controlled runoff process are controlled/ limited by intensive properties such as rainfall intensity and/or infiltrability. This implies that proper normalization depends on the nature of the runoff processes, and suitable estimators for intensity controlled runoff are not straight forward to estimate at the lower mesoscale. We will revise the introduction of the manuscript accordingly and particularly reformulate the question 2 in the specified sense.

With low frequent data we mean that a daily and/or even hourly sample is to coarse to sample for instance fast convective precipitation events, which might trigger intensity controlled runoff formation. Additionally, flood routing in the river net implies dispersive smoothing of the sharp peaks, which implies dampening of the high frequent components within hydrographs.

RC: Why "particularly at the lower mesoscale."? Authors reply: The lower mesoscale refers to catchment sizes of few square kilometers to about 100 km$^2$, which are due to Dooge (1986) systems of organized complexity. We selected this scale because our understanding of the interplay of how catchment structure and its state jointly control

runoff generation mechanisms is rather incomplete, and it is already large enough to pool 100 catchments into the sample (though only 22 were used here). Moreover, at this scale routing effects are still small, as for instance Robinson et al., (1995) pointed out that catchments up to 20 km$^2$ are still hillslope dominated.

RC: What is the coherence of Q1-Q3 beyond "testing dimensionless measures to discriminate differences in runoff generation"? Authors reply: We will reformulate these questions to better reflect that these normalized measures (particularly the question how to normalize) depend on a) the selected time scale (seasonal or event) and b) at the event scale on the nature of the runoff formation processes. The double mass curves we use at the seasonal scale are in fact pretty similar to common practice in soil physics to plot tracer breakthrough against cumulated irrigation (and not against time). Contrary to soil physics we have to forms of water release (evaporation and stream flow) and the proposed double mass curves are particularly suited to separate regimes where either the one or the other is dominating. Note that particularly the summer regimes are easily explained by the temperature index model proposed by Menzel et al., (2003), which explains onset of the vegetation phase. At the event scale we may use different storage estimators and check which of those explains most of the observed runoff coefficients, when assuming storage control as dominant. Or we face the problem how to detect, characterize and normalize intensity controlled runoff formation.

RC: The paper presents the findings as "generally applicable for meso-scale catchments". This is an extremely bold statement since only 22 catchments in a very small part of the world are used. Authors reply: This is a misunderstanding, we did not mean that our findings are generalizable to all catchments in the world, with respect to for instance the temperature index being a good variable to explain regime shifts. We propose that the suite of measures is applicable to mesoscale catchments of humid environments were they can applied as a starting point to learn about the interplay of structure, state and runoff response. We will clarify this in the revised manuscript.

We again sincerely thank the reviewer for the helpful comments, which are certainly helpful for improving our study and thank for the note that the reviewer is sure that we have interesting findings to share. We will restructure the manuscript in line with most suggestions and technical details recommended by the reviewer.

References

Dooge, J.C.I., 1986. Looking for hydrological laws. Water Resour. Res. 22, 46S–58S.

Menzel, A., Jakobi, G., Ahas, R., Scheifinger, H., Estrella, N., 2003. Variations of the climatological growing season (1951–2000) in Germany compared with other countries. Int. J. Climatol. 23, 793–812. doi:10.1002/joc.915

Robinson, J.S., Sivapalan, M., Snell, J.D., 1995. On the relative roles of hillslope processes, channel routing, and network geomorphology in the hydrologic response of natural catchments. Water Resour. Res. 31, 3089–3101. doi:10.1029/95WR01948

Zehe, E., Ehret, U., Pfister, L., Blume, T., Schröder, B., Westhoff, M., Jackisch, C., Schymanski, S.J., Weiler, M., Schulz, K., Allroggen, N., Tronicke, J., van Schaik, L., Dietrich, P., Scherer, U., Eccard, J., Wulfmeyer, V., Kleidon, A., 2014. HESS Opinions: From response units to functional units: a thermodynamic reinterpretation of the HRU concept to link spatial organization and functioning of intermediate scale catchments. Hydrol. Earth Syst. Sci. 18, 4635–4655. doi:10.5194/hess-18-4635-2014

Zehe, E., Ehret, U., Blume, T., Kleidon, A., Scherer, U., Westhoff, M., 2013. A thermodynamic approach to link self-organization, preferential flow and rainfall–runoff behaviour. Hydrol. Earth Syst. Sci. 17, 4297–4322. doi:10.5194/hess-17-4297-2013

---

## Referee Comment (RC2) · Anonymous Referee #2 · 12 Apr 2016

This paper discusses catchment functions with particular focuses on the storage dynamics. It deals with various important aspects with a number of related references, and the scope of the paper fits to HESS.

However, current manuscript shows so many different components and unfortunately they are not well interlinked to achieve the authors' original challenge on explore the interplay between state, structure and runoff behavior.

Although one of the main focuses is on the normalized dimensionless storage predictors, as primarily described in the method, the manuscript also touches upon other things including the relationship between topography and runoff ratios, temporal sampling frequency, triple mass curves etc. As a result, it is currently very difficult to understand what the authors would like to solve or propose in this single paper and the significance of the manuscript becomes unclear.

Furthermore, the logic of the evaluation for the three storage predictors must be well defined. Obviously different predictors represent different properties, yet they are all related to the catchment storage. Therefore well understanding of the predictors' characteristics is very important in qualitative way. However, what confuses me is that the authors tend to say the predictor showing the higher correlation to the compared indices is the best. Related statements appear many times, for example on L784 in P.24. Please describe clearly your logic on the evaluation of the predictors.

Secondly, the rationale of the normalization is unclear. The values can be easily converted to be non-dimensional, but it is effective only if you can normalize them in a physically meaning manner. In the current manuscript, all of the result figures show the relationship between the predictors and the compared variables at each catchment and its evaluation basically conducted based on the rank correlation. If this is the purpose, I do not see any necessity on the normalization.

Moreover, the equation (3) normalizes the average discharge volume divided by soil porosity. What does this mean physically? Also the equation (2) sums up the difference between (P - E) whose total values vary significantly depending on the duration of the summing up. If so, how can you convince it was successfully normalized by the porosity? Why can it be better than the original values? The same concern is applied to the equation (4) also, especially Ks from a soil map can vary significantly by some orders with large uncertainty.

In addition to the major above major concerns, I have the following minor comments..

1. P1. L9: the meaning of "extensive/additive" and "intensive/non-additive" is unclear.

2. P1. L14: "the latter case" is unclear to specify the listed three items.

3. P1. L20: what is "proposed non-additive response measure"?

4. P2. L24 does the summary of the conclusion starting from L24 actually correspond to the original objective of the study?

5. P9 L294 please describe the method to separate event quick flow and base flow.

6. P12 L394 Larsim -> LARSIM to be consistent to L389

7. P15 L487 explain what is "power model exponent"

7. P16 L527 It is unclear if "CR_E and theta are often pretty linear, whereas that between CR_E and dS indicates threshold behavior" according to the figure.

8. P17 L545-550 This part is not clear.
* * *

---

## Referee Comment (RC3) · Anonymous Referee #3 · 16 Apr 2016

This is an interesting paper that explores the underlying functional behavior of catchments in terms of dimensionless quantities that describe catchment processing/transmission of water. It is clear that the authors have done a lot of work – timeseries analysis, modeling, etc – to complete this study. I particularly enjoyed reading it! However, I found it very difficult to follow details of the analysis throughout, and have questions related to uncertainty and how it may propagate in a study such as this.

A few questions for them: 1. While I generally enjoyed the paper, I found the organization of section 3 very difficult to follow. What data do you use to compute your statistics? Many different data are mentioned in this section including met data (relative humidity, etc) and modeled data (ET, etc). I am guessing some met data are

mentioned only because they were needed for modeling, while other products were used directly to compute statistics. It would be helpful if you directly state which data are used to compute your statistics and which are used for LARSIM. I think you were trying to be inclusive, but the generality here leaves me a little confused. For instance, what data are included in hydro-meteorological time series? 2. Along these same lines, you mention catchment characteristics data in section 3.1, but do not direct readers to Tables 1 and 2, or mention which catchment characteristics you derive or why you derive them. Brevity is good, but I found that without introducing them, when you mentioned derivatives from these datasets in results and discussion I was a little lost. 3. I guess I see the requirement of modeling (to obtain ET) as a potential source of added uncertainty in this analysis. While I see that the modeling results are published in a second paper, and I recognize you do not have to defend these results in this new paper (and as well that ET is an important part of a catchment functioning analysis), I do think it is important that you acknowledge some of the uncertainty that comes from using a derivative of a model in this type of analysis. I think our community does a good job acknowledging potential sources of uncertainty from modeling and problems related to equifinality, but I also think we run the risk of incorrect interpretation when we do not directly include this uncertainty in analyses such as the one presented in this paper. For instance, if your input data to your model was assumed to have an error of +/- 5 or 10%, and this error was propagated you're your ET signal, would this change your conclusions? It might be worth adding a short discussion of potential limitations in such an analysis. I'd be interested to see if you come to similar conclusions if one or two of the sites are located close to an Eddy-Covariance tower, and if those measured values were used in this analysis (recognizing as well that ET from Eddy-Covariance is a derivative data product with error). 4. In the introduction to the paper and in the discussion, I felt like there were two sources of literature not acknowledged. Again, brevity is important, but this work reads as very similar to the literature on runoff generation and the literature on catchment classification. I noticed the absence of some seminal/recent papers from both sources that could potentially bolster your introduction and conclusions. 5. I completely understand needing to use abbreviations in results/discussion, but I found it again very difficult to link your acronyms back to their description in Section 2 of the paper. A table linking acronyms, descriptors, and their importance in determining catchment functioning would organize the many different pieces. 6. You introduce the ideas of additive vs. non-additive in the abstract, but this does not follow throughout the paper. Consider removing, reframing. 7. What makes a dimensionless quantity better than a dimensional quantity? How is what you have framed different from other catchment classification studies? Why should we normalize measures others have used?

Minor comments: -A minor point, but consider limiting the use of the word "essentially" – there were a few sentences when it was used twice! I think it detracts from the quality of your point Line 51+: grammar Line 79+: grammar – relative? Line 674: full reference missing

---

## Editor Comment (EC1) · FF Fenicia (Editor) · 18 May 2016

I think the work is potentially interesting, and within the scope of the journal. However, the paper needs to improve in terms of the presentation of the material. In addition, the paper attempts to cover a lot of ground, which requires either a better synthesis, or a selection of topics, deferring other material to a separate work. I recommend the authors to address the reviewers criticisms, which implies a major revision of the paper. In addition to the reviewers comments, I have the following suggestions.

1. One of the purposes of the paper is to introduce dimensionless indices to characterize the catchment structure and function. However, the units of most of the quantities that goes into these indices are not reported. Using common sense for the units of

the various quantities, these indices do not appear to be dimensionless. For example see Equation 1. AC and eFC are usually dimensionless (e.g. Reynolds et al., Geoderma, 2002), whereas P, E and Q are usually in mm/time, meaning that dS* is not dimensionless.

2. Is there a clear advantage in using scaled quantities, such as the ones proposed, over using not scaled quantities? Would it be possible to show it? Do scaled quantities lead to better fits than not scaled quantities?

3. Clarify the difference between dimensionless and normalized

4. State and structure are terms that are usually applied in the context of models. Here the authors apply these terms in the context of catchments, i.e. of natural systems. The meaning of this terms needs to be clarified at the beginning of the paper.

5. The objectives are a key part of the paper. I think they are not very clear. What does "feasible" mean in objective 1? Please elaborate and clarify objective 2. Objective 3 would benefit from a clarification of the terminology, e.g. why catchment structure does not include ecology. Results and conclusions should demonstrate how the objectives are reached (i.e. not leave it to the reader to find out).

6. The paper talks about "model based estimates of evapotranspiration" without explaining how these estimates are obtained (some details are given, but much later in the paper). Restructure to clarify and motivate this choice.

7. The catchments are classified based on physiography, but the conclusions say nothing about insights on catchment behaviour. Which catchment characteristics control hydrologic response at that scale?

8. Can some of the tables be converted to figures?

---

## Author Comment (AC2) · 1 Jul 2016

**Final response**

We sincerely thank the three anonymous referees for their detailed comments, as well as the editor for giving us the opportunity to revise our manuscript.

Here we provide our final response to all major comments made by the referees and the editor. Since our manuscript will be subject to major revisions we do not provide a point-by-point list, but rather a general reply to what we have identified as the most important issues related to our initial contribution.

For RC#1, but also for RC#2, RC#3 and the editor, the presentation of the material was a key point of concern. Among all referees was a consensus on the lack of a logical and smooth structure, ultimately making the manuscript difficult to read. We can see this point (now) and agree that several sections should be re-written and/or re-structured. This includes particularly the abstract, introduction & objectives, the part on "conceptual framework and candidate diagnostics", as well as the conclusions. Although we agree with RC1 that the paper could be shortened by such a revision, the comments also suggest that several aspects need to be clarified and explained in more detail. This concerns in particular the rationale of the normalization, the logic of the storage predictors and the whole concept of "intensity controlled" runoff production. We understand almost all of the related points of criticism and will revise our manuscript according to the suggestions of the referees/editor. However, we suspect that the clarification of these issues might further extend the paper – which is already very long. For this reason, we decided to split our contribution into a companion study. We suggest that the initially proposed manuscript becomes part 1 and to defer certain elements into a second manuscript, which we will submit separately. We intend to organize the companion study along the following avenues:

> Manuscript 1) Exploring the interplay between state, structure and runoff behavior of lower mesoscale catchments I: unraveling abiotic and biotic controls on the seasonal water balance.

> Manuscript 2) Exploring the interplay between state, structure and runoff behavior of lower mesoscale catchments II: storage versus intensity controlled event runoff production.

The overarching goal of our contribution is to better understand the interplay between state, structure and runoff behavior in lower mesoscale catchments. By distinguishing between seasonal and event runoff production we study catchment runoff dynamics at different time scales. At event scale we furthermore differentiate according to the nature of the runoff formation process. Our key intention is to isolate multiple impacts on runoff production, i.e. to separately study the impact of catchment structure, moisture state or that of biotic controls. To discriminate among the impacts of the rainfall forcing and that of the structural (terrestrial) properties of the catchment we use normalization. In essence we hypothesise that normalized state-response and forcing-response plots are suitable diagnostics to discriminate differences in catchment runoff production at different time scales. We further differentiate this general hypothesis (H) within the two contributions and test it through complementary research questions (Q):

**Paper 1: Abiotic and biotic controls on the seasonal water balance**

> **H1:** Normalized forcing-response plots are suitable diagnostics to discriminate biotic and abiotic controls on seasonal catchment runoff production.

> **Q1:** How to evaluate and normalize seasonal runoff production, so that the impact of biotic and abiotic controls can be evaluated independent from meteorological forcing and/or the structural setup of the catchment?

> **Q2:** Which structural catchment characteristics explain the differences in seasonal runoff production during the dormant season and do any of them operate in groups?

**Paper 2: Storage versus intensity controlled event runoff production**

> **H2:** Normalized state-response and forcing-response plots can discriminate differences in catchment runoff production at the event time scale.

> **Q1:** How to estimate and evaluate the impact of different catchment water storages on event runoff production?

> **Q2:** How to normalize both storage and response to (i) consider the site specific structural setup of the catchment and (ii) detect similarities in event scale runoff production in inter-comparison studies?

> **Q3:** Is it possible to detect evidence for intensity controlled runoff formation, which refers to the activation of rapid flow paths, such as surface runoff or preferential flow, based upon commonly available (hourly aggregated) data of mesoscale catchments

In addition to our reply to the anonymous referee # 1 we provide answers to the major aspects raised by the reviewers/editor in the following:

**What is the rationale of normalization?**

To normalize means to relate a variable of interest to a reference, or to express it in terms of the latter. Such a normalization allows to separate multiple impacts on a single variable and facilitates studying the importance of different properties on a single quantity. Normalization hence allows to compare different sites, which makes it a promising tool for comparative analyses. Well-known and widely used examples of normalized quantities include specific discharge, typically expressed in [length/time] where the impact of the catchment size [length x length] is separated from the discharge measurement [volume/time], or runoff coefficients [-] where the response, i.e. again specific discharge [length/time] is normalized such that it is independent from the forcing, i.e. specific rainfall [length/time]. Normalized quantities are often used as "diagnostic" variables to study runoff generation (e.g. Merz et al., 2006 or Graeff et al., 2012).

In line with these studies we propose that the analysis of normalized runoff surrogates allows to separate the impact of the rainfall forcing from that of the terrestrial influences, i.e. catchment state and structure. The latter includes both abiotic and biotic components. In essence, we suggest that the inherent "functional behavior" of a catchment with respect to runoff generation can be described by normalized state-response and/or forcing-response diagrams.

In our study, we analyze runoff generation at different time scales. Specifically, we distinguish between runoff production on the seasonal and on the event time scale. On the latter we furthermore differentiate according to the nature of the runoff formation process. Accordingly, we use different normalization schemes.

*At seasonal scale*: The double mass curves (DMC), i.e. normalized forcing-response plots, which we use at the seasonal scale are in fact pretty similar to common practice in soil physics where tracer breakthrough is plotted against cumulated irrigation (and not against time) to study transport and adsorption. Contrary to soil physics we do however have two forms of water release in catchment hydrology (evaporation and stream flow) and the proposed double mass curves are particularly suited to separate regimes where either the one or the other is dominating. In the current version of the manuscript the DMCs are normalized with total precipitation. This has the advantage that the abscissa is always scaled from 0 to 1 which facilitates the comparison of different sites and years. It has however the weakness that the same ordinate values do not reflect the same mass input. An alternative would be to normalize with storage volume (despite that the uncertainty is high), which expresses mass input in terms of storage volume. A further alternative would be to use a data-driven estimate for the maximum potential evaporation such as net solar radiation divided by the latent heat of vaporization (assuming the entire incoming energy would be consumed for evaporation). This would efficiently separate cold from warm years. We will further elaborate and test the potential of the different normalization schemes for describing the seasonal water balance in the revised manuscript. Furthermore, we will include common-practice signatures and relate insights obtained from the DMCs to results obtained from flow duration curves. The latter are widely used as hydrological signatures in similarity studies.

*At event scale*: at event scale we distinguish between "capacity" controlled runoff production and "intensity" controlled runoff production. The former relates to mechanisms where the relative amount of water which is stored in the control volume dominates the response. We expect that capacity controlled runoff production, meaning that streamflow monotonically depends on storage amount is the most important mechanism in many, particularly humid environments. The key challenges in this context are a) to characterize different storage compartments, which is done using the different storage estimators and b) to estimate their relative content, which is done by normalizing the storage estimator using a surrogate for the storage capacity in the subsurface. Intensive runoff production refers to the activation of rapid flow paths such as surface runoff or preferential flow. The activation of these flow paths requires an intensive, typically convective rainfall forcing (Beven and Germann, 2013) and is largely independent from the amount of water which is stored in the subsurface. (Relative) storage is hence a poor predictor for intensity controlled runoff production and it is clear that the detection of these mechanisms by means of normalization requires to consider the nature of the process. The use of (storage) capacities, which behave additive, is less appropriate in this case. Intensities, which do not behave additively, actually are the much more relevant properties in that event. Estimating all relevant intensities, or more generally the "intensive state variables" is however not straightforward. It would require structural information on the rapid flow paths, i.e. related conductivities, knowledge on

the intensity of the forcing and on that of the response, both in a very high spatio-temporal resolution. Unfortunately, the former are not available and the latter, i.e. discharge, is a convolute of different distributed runoff generation processes, concentration and routing. For this reason we suggest to focus on high intensive rainfall events and to consider associated high intensive runoff responses as evidence for the activation of rapid flow paths. To detect such processes within rainfall-runoff events and based upon data at hourly resolution, which are poorly resolved in this context, we propose to relate the temporal derivatives of rainfall and runoff. This implies to "edge-filter" the time series and to normalize the temporal derivative of discharge using the temporal derivative of precipitation. This emphasizes rapid changes in intensities much better than the observed values.

We recognize that in the first version of our manuscript we did not concisely differentiate between "normalized" and "dimensionless" variables. While many of our normalized quantities are dimensionless, the latter is of course not a prerequisite for the former. We therefore thank the editor for pointing out that both terms must not be mistaken and clearly differentiated. We will clarify this in the revised version of our manuscript.

**What is the logic of defining and evaluating the different storage predictors?**

The importance of storage on runoff production is beyond question. Characterizing and normalizing storage is however difficult, as elaborated in section 1.3. Basically, we intend to 1) assess the importance of different sub-surface storage compartments on event runoff production (and associated spatial patterns) and 2) to detect similarities in storage capacity controlled runoff production among different sites by means of an inter-comparison study. To characterize storage we selected total active storage dS (Sayama et al. 2011) (Eq. 1) which we associated mainly with deeper storage compartments, and theta (Eq. 2) as a surrogate for the near-surface storage. Pre-event discharge (Eq. 3) was included as it proved to be a valuable estimator for the bulk catchment moisture state (Graeff et al. 2012), although it cannot be attributed clearly to either the near-surface or deeper storage compartments.

To evaluate the impact of the different storage compartments on runoff response we correlate storage estimators and event runoff coefficients. The number of significant correlations gives insights into the overall importance of the different storage surrogates, i.e. the importance of the different sub-surface storage compartments, within our data set. Spatial patterns provide information on regional differences in the importance of the storage compartments. The site-specific normalization is necessary to judge the relative importance of the storage estimators which allows a meaningful comparison of different catchments.

Currently, the evaluation focuses on general aspects of the proposed method. Therefore, we mainly provided results for individual catchments and not for the inter-comparison of different sites like it is done in Fig. 5, bottom right. Splitting the manuscript will allow us to provide more results and to explain them in more detail. Thanks also to RC#2 for pointing out that there is no "best" storage estimator as different predictors represent different properties, yet they are all related to catchment storage. We will clarify this in the revised version of the manuscript. Therein, we will also detail on our rationale for defining the time of integration/summation in the proposed storage measures (Eq. 1-3). It is of course true that the time of integration/summation

introduces a high degree of subjectivity and thus, uncertainty. This has already been pointed out by Heggen (2001) and Graeff et al. (2012).

In the revised manuscript we will also provide the units of all quantities which we use in our analyses and show that Eq. 1-3 are dimensionless. We thank the editor for highlighting that this aspect need clarification.

**Which data have been used for which purpose and what is the role of the hydrological model?**

The fact that observables are generally less uncertain than model outputs motivated us to employ data-driven signatures whenever possible. Data-driven means that our signatures are based upon observables and not upon model outputs. This applies for all signatures, except for those which include ET estimates. ET was calculated based upon Penman-Monteith using a water budget model (LARSIM). Calculating ET required standard hydro-meterological time series, i.e. observed station data of radiation, wind speed, humidity, temperature, etc. We use the same hydrological model for all catchments to ensure that the ET estimates are calculated and interpolated in a consistent way.

In the revised version of the manuscript we will clarify the use of the different data and explain which of them were used for modelling and which of them were directly used to derive the different signatures. Following the suggestions of RC#2 and RC#3 will also comment on the associated uncertainty.

**References**

Beven, K., Germann, P., 2013. Macropores and water flow in soils revisited. Water Resour. Res. 49, 3071–3092. doi:10.1002/wrcr.20156

BGR, 1995. Bodenübersichtskarte von Deutschland 1:1,000,000 (BÜK1000). Federal Institute for Geosciences and Natural Resources (BGR).

Merz, R., Blöschl, G., Parajka, J., 2006. Spatio-temporal variability of event runoff coefficients. J. Hydrol. 331, 591–604. doi:10.1016/j.jhydrol.2006.06.008

Graeff, T., Zehe, E., Blume, T., Francke, T., Schröder, B., 2012. Predicting event response in a nested catchment with generalized linear models and a distributed watershed model. Hydrol. Process. 26, 3749–3769. doi:10.1002/hyp.8463

Heggen, R.J., 2001. Normalized Antecedent Precipitation Index. J. Hydrol. Eng. 6, 377–381. doi:10.1061/(ASCE)1084-0699(2001)6:5(377)

Sayama, T., McDonnell, J.J., Dhakal, A., Sullivan, K., 2011. How much water can a watershed store? Hydrol. Process. 25, 3899–3908. doi:10.1002/hyp.8288

---

## Author Response (AR1)

**Dear Editor,**

let me first of all thank you for providing us extra time to reply to the reviews, which we found most helpful for improving our manuscript.

According to the suggestions of the three anonymous referees the manuscript "Exploring the interplay between state, structure and runoff behaviour of lower mesoscale catchments" was subject major revisions. The most important aspect in this respect was that we splitted the study and deferred its contents into two separate manuscripts: The first part deals with seasonal runoff formation and the water balance, the second part with runoff generation at the event scale. To this end we finished the revision of the first part. A revised version on event scale runoff formation signatures will be submitted separately. In line with our responses to the reviewers we revised our manuscript as follows:

- To clarify the (new) scope of the article we changed the title into: "Unraveling abiotic and biotic controls on the seasonal water balance using data-driven dimensionless diagnostics".

- Hypothesis and research questions were revised to clarify the overall goal of the manuscript.

- The manuscript was re-structured and several chapters were entirely re-written. This includes particularly abstract, introduction, and discussion and conclusions. The results on seasonal runoff formation are in essence the same but were complemented with additional findings to explain the proposed method in more detail. Compared to the first submission the manuscript was shortened by 15 pages.

- A key aspect we address in our paper is the derivation of dimensionless fluxes which ensure the comparability of the signatures in space and time. We clarify this point in abstract and conclusions and provide details on two different approaches in section 2.2.

All comments raised by the referees and by the editor were considered. In the following we provide brief revision notes to all major aspects.

#########################################################################
**Editor Comments**

–

1. ...the units of most of the quantities that goes into these indices are not reported. Using common sense for the units of 1. AC and eFC are usually dimensionless, whereas P, E and Q are usually in mm/time, meaning that dS* is not dimensionless.

**Authors reply:** in the revised manuscript all units are reported. We furthermore explain that AC and eFC, both provided in (mm/dm), are multiplied by root zone depth (dm) and hence converted to (mm).

–

2. Is there a clear advantage in using scaled quantities, such as the ones proposed, over using not

scaled quantities? Would it be possible to show it? Do scaled quantities lead to better fits than not scaled quantities?

**Authors reply:** We employ dimensionless diagnostics to ensure the comparability of the signatures in space and time - not to obtain better fits. Comparability is vital for intercomparsion studies. We explain this in more detail in the revised version of the manuscript.

–

3. Clarify the difference between dimensionless and normalized

**Authors reply:** We apologize for not being precise. In the revised version we only use the term "dimensionless" and avoid any use of "normalized".

–

4. State and structure are terms that are usually applied in the context of models. Here the authors apply these terms in the context of catchments, i.e. of natural systems. The meaning of this terms needs to be clarified at the beginning of the paper.

**Authors reply:** We clarify this issue in the introduction (P3 L15).

–

5. The objectives are a key part of the paper. I think they are not very clear. What does "feasible" mean in objective 1? Please elaborate and clarify objective 2. Objective 3 would benefit from a clarification of the terminology, e.g. why catchment structure does not include ecology. Results and conclusions should demonstrate how the objectives are reached (i.e. not leave it to the reader to find out).

**Authors reply:** We reformulated the objective and adapted the research questions according to the new scope of the paper.

–

6. The paper talks about "model based estimates of evapotranspiration" without explaining how these estimates are obtained (some details are given, but much later in the paper). Restructure to clarify and motivate this choice.

**Authors reply:** The method section was restructured. Information on the site and the available data is now introduced before we detail on the applied methods.

–

7. The catchments are classified based on physiography, but the conclusions say nothing about insights on catchment behaviour. Which catchment characteristics control hydrologic response at that scale?

**Authors reply:** The conclusions were revised and include several statements concerning the insights we obtained on catchment behaviour. Among others, these include:

- P18 L27: … temperature data prove to be good predictors for $CR_S$, independent from the physiographic and climatological conditions represented by our data set.

- P19 L11: ...the product of the topographic gradient and the saturated hydraulic conductivity is an important predictor for the average winter runoff coefficients while the two variables alone are not.

- And more general P20 L6: dimensionless double mass curves in combination and ecological temperature index is well suited to unravel biotic and abiotic controls on seasonal runoff formation as long as runoff formation does monotonously increase with storage.

–

8. Can some of the tables be converted to figures?

**Authors reply:** The revised paper was shortened by 15 pages. Only 7 out of 11 figures and 4 out of 6 tables are included in the revised version of the paper. For this reason we decided not to convert any tables into figures.

###############################################################################
**Reply to anonymous Referee #1**

–

1) I cannot find a clear main novel contribution of this paper from either the abstract or the conclusion. This may be challenging to formulate as the paper tries to understand catchment similarity for different time-scales and processes, but I do expect that there are some overarching goals.

**Authors reply:** Abstract and conclusions were re-written.

–

2) The abstract is a mess: first you state a very broad and poorly defined problem, sub- sequently you state many details of the analysis that are not informative is the problem is still unclear. For example how is the reader supposed to know what you mean by "extensive/additive" and "intensive/non-additive". After you introduced the methods, your results focus on differences in functioning at different time-scales. Why didn't you introduce that you are looking at different timescales in your problem statement or methods? Your results do not clearly link to the introduction and methods part of your paper, and read like things you ad-hoc found and listed. You start stating results "Our dimensionless signatures evidently detect. . ." before the reader knows what data this comes from. This seems like an illogical and confusion order to me. Your conclusions feel like a random list of unconnected findings. Do you really need so many words for your abstract? I think you can (and should) be more to the point.

**Authors reply:** The  abstract was re-written. Terms such as "extensive/additive" were deleted.

–

3) The introduction section lacks a logical build-up to a problem statement. Although I acknowledge that the introduction is giving a relevant overview of many challenges we face in

concisely characterizing how catchments function, it reads like a long list of problems, rather than a structured introduction working defining a clear problem and working towards a clear goal.

**Authors reply:** The introduction was revised according to this suggestion and build-up towards a clear goal. We hope that the new version is more clear.

_

4) Are you sure that "Question 2: Can we detect intensity controlled runoff formation essentially a high frequency process based on low frequency data?" is a clearly defined question for the (ignorant) reader (like me).?

**Authors reply:** Question 2 was reformulated. A more precise definition of " intensity controlled" is provided (P8 L32).

_

5) What is the coherence of Q1-Q3 beyond "testing dimensionless measures to discriminate differences in runoff generation"? If there is coherence, please formulate it such that this is clear. If there is no coherence, why would you address these three questions within a single manuscript?

**Authors reply:** Due to the splitting of the manuscript Q1-Q3 were reformulated. This issue is not relevant any more.

_

6) The section "Conceptual framework and candidate diagnostics" needs to be improved. Although you list requirements of the paper in the different subsection the writing is messy. I think you should be much more to the point in clearly stating the requirements. Problems that I came across and a. "Requirements of functional diagnostics" is mainly an overview of your perception of how the hydrology of different landscapes function. Right now the reader reads one sentence about the "requirements" but are confused by the end of the section 2.1 what the purpose of this section is. b. The requirements of "Normalization of states and re- sponse measures" and ""Coherence and quality of integral storage measures" should be concisely written and to the point. c. I understand that you choose the metrics in section 2.2 based on all the requirements you have set before, but when you introduce them in this section I expect that the rationale of choosing them is clear. Currently it is not. For example "Lastly we use a normalized specific pre-event discharge (Q ) av- eraged across the last seven days:" is just an announcement, but leaves it completely unclear where the rationale to choose this metric comes from. The introduction of other metrics suffers from similar issues.

**Authors reply:** The section "Conceptual framework and candidate diagnostics" was entirely re-structured and re-written.

_

7) Is it logical that you provide the details of your study area only after you have discussed all the metric you use? I presume these metrics are largely based on your perception how the study catchment function. Therefor I think it would be more logical to switch their order.

**Authors reply:** We switched the order of the sections.

–

8) There should be a story in your results, rather than that it is a long list of ad-hoc results you have obtained. For example you begin with the statement that "During low flow conditions, the storage estimators dS and are in most cases linearly independent." And subsequently you list the associate statistics. The logic in your story is impossible to follow (for me). Listing the how the storage metrics link to flow characteristics is fine, but please make it more read like a story.

**Authors reply:** We re-structured the results to clarify the storyline. We hope that the revised version is easier to read.

–

9) The paper presents the findings as "generally applicable for meso-scale catchments". This is an extremely bold statement since only 22 catchments in a very small part of the world are used.

**Authors reply:** This was a misunderstanding that we clarified in the revised version. In P16 L16 we "conclude that season-specific dDMCs are a well suited fingerprint for characterizing seasonal runoff formation in meso-scale catchments and that dDCMs are suited for intercomparison studies."

–

10) The paper is uses many words, while I am not against long papers, it does feel that with 30% less words the story can be told too. The current version of the manuscript is not clear enough to allow me to properly evaluate the results obtained in the paper. Therefor I suggest that this paper can only be considered for HESS after rigorous rewriting and restructuring of the paper has been done. I am sure that interesting new findings are available within the results, but at this stage I can not properly evaluate this.

**Authors reply:** The revised version was shortened by 15 pages.

################################################################################
**reply to referee #2**

Although one of the main focuses is on the normalized dimensionless storage predictors, as primarily described in the method, the manuscript also touches upon other things including the relationship between topography and runoff ratios, temporal sampling frequency, triple mass curves etc. As a result, it is currently very difficult to understand what the authors would like to solve or propose in this single paper and the significance of the manuscript becomes unclear.

**Authors reply:** For this reason we splitted the manuscript. This study focuses on runoff formation at the seasonal and annual time scale. The methods for the event scale will be presented in a follow-up study.

–

Furthermore, the logic of the evaluation for the three storage predictors must be well defined. Obviously different predictors represent different properties, yet they are all related to the catchment storage. Therefore well understanding of the predictors' char- acteristics is very important in qualitative way. However, what confuses me is that the authors tend to say the

predictor showing the higher correlation to the compared in- dices is the best. Related statements appear many times, for example on L784 in P.24. Please describe clearly your logic on the evaluation of the predictors.

**Authors reply:** This aspect refers to the event scale measures. In this manuscript we focus on the seasonal time scale. The issues are hence not relevant any more.

–

Secondly, the rationale of the normalization is unclear. The values can be easily converted to be non-dimensional, but it is effective only if you can normalize them in a physically meaning manner. In the current manuscript, all of the result figures show the relationship between the predictors and the compared variables at each catchment and its evaluation basically conducted based on the rank correlation. If this is the purpose, I do not see any necessity on the normalization.

**Authors reply:** We clarify this aspect and state that dimensionless approaches are required to ensure comparability of signatures in space and time. The derivation of dimensionless quantities is a key aspect of our paper. We discuss this aspect in detail in section 2.2.

–

Moreover, the equation (3) normalizes the average discharge volume divided by soil porosity. What does this mean physically? Also the equation (2) sums up the difference between (P - E) whose total values vary significantly depending on the duration of the summing up. If so, how can you convince it was successfully normalized by the porosity? Why can it be better than the original values? The same concern is applied to the equation (4) also, especially Ks from a soil map can vary significantly by some orders with large uncertainty.

**Authors reply:** The key to obtain scale invariant dimensionless quantities is to divide a state variable of interest – for instance a force, velocity or length by a characteristic quantity of the system (Blöschl and Sivapalan, 1995). A popular example is the Reynolds number in fluid mechanics which relates inertial forces to viscous forces. It is defined as flow velocity times a characteristic length divided by the dynamic viscosity of the fluid, which is suited to compare turbulence in open channel flow independent of the channel width. The best known example of a dimensionless response measure in hydrology is the runoff coefficient (CR) (-) defined as specific discharge (L/T) over specific rainfall (L/T). The latter is often used as "diagnostic" variable to detect scale invariant differences in generation of runoff volume (Merz et al., 2006, Graeff et al. 2012).

In the revised version of the manuscript we explain two versions to obtain dimensionless double mass curves. We show that the kind of scaling strongly influences the diagnostic potential of the signatures.

We are well aware that data on soil porosity are highly uncertain, particular at the catchment scale, but this is what is operationally available in Southern Germany.

###############################################################################
**reply to reviewer 3**

1. While I generally enjoyed the paper, I found the organization of section 3 very difficult to follow. What data do you use to compute your statistics? Many different data are mentioned in this section including met data (rel- ative humidity, etc) and modeled data (ET, etc). I am guessing some met data are mentioned only because they were needed for modeling, while other products were used directly to compute statistics. It would be helpful if you directly state which data are used to compute your statistics and which are used for LARSIM. I think you were trying to be inclusive, but the generality here leaves me a little confused. For instance, what data are included in hydro-meteorological time series?

**Authors reply:** Section 3 was entirely re-written and re-structured.

–

2. Along the same lines, you mention catchment characteristics data in section 3.1, but do not direct readers to Tables 1 and 2, or mention which catchment characteristics you derive or why you derive them. Brevity is good, but I found that without introducing them, when you mentioned derivatives from these datasets in results and discussion I was a little lost.

**Authors reply:** In the revised version we introduce the tables and describe how the different properties are derived.

–

3. I guess I see the requirement of modeling (to obtain ET) as a potential source of added uncertainty in this analysis. While I see that the modeling results are published in a second paper, and I recognize you do not have to defend these results in this new paper (and as well that ET is an important part of a catchment functioning analysis), I do think it is important that you acknowledge some of the uncertainty that comes from using a derivative of a model in this type of analysis. I think our community does a good job acknowledging potential sources of uncertainty from modeling and problems related to equifinality, but I also think we run the risk of incorrect interpretation when we do not directly include this uncertainty in analyses such as the one presented in this paper. For instance, if your input data to your model was assumed to have an error of +/- 5 or 10%, and this error was propagated you're your ET signal, would this change your conclusions? It might be worth adding a short discussion of potential limitations in such an analysis. I'd be interested to see if you come to similar conclusions if one or two of the sites are located close to an Eddy-Covariance tower, and if those measured values were used in this analysis (recognizing as well that ET from Eddy-Covariance is a derivative data product with error).

**Authors reply:** Due to the splitting of the manuscript the focus is now on data-driven diagnostics, that means we only use observables. ET is hence not required any more. Accordingly, uncertainties are less important - though of course not negligible. However, to keep the manuscript as concise as possible we decided not to discuss uncertainty issues in our paper.

–

4. In the introduction to the paper and in the discussion, I felt like there were two sources of literature not acknowledged. Again, brevity is important, but this work reads as very similar to the literature on runoff generation and the literature on catchment classification. I noticed the absence

of some seminal/recent papers from both sources that could potentially bolster your introduc- tion and conclusions.

**Authors reply:** In the revised version of the paper we provide literature on runoff generation and catchment classification.

–

5. I completely understand needing to use abbreviations in results/discussion, but I found it again very difficult to link your acronyms back to their description in Section 2 of the paper. A table linking acronyms, descriptors, and their importance in determining catchment functioning would organize the many different pieces.

**Authors reply:** Due to the splitting of the manuscript the paper is clearer and we use less abbreviations and acronyms. For this reason we do not provide a table that links acronyms and descriptors.

–

6. You introduce the ideas of additive vs. non-additive in the abstract, but this does not follow throughout the paper. Consider removing, reframing.

**Authors reply:** These terms have been removed.

7. What makes a dimensionless quantity better than a dimensional quantity? How is what you have framed different from other catchment classification studies? Why should we normalize measures others have used?

**Authors reply:** Please see our answer to the last question of referee #2 and section 2.2 in the revised version of the paper.

---

## Editor Decision (ED1)

I found the paper significantly improved in the text and structure, however there may be some problems with the analysis, which I would like you to check before I proceed further. Below my notes while I was reading the paper.

Abstract.

Page1, Line 1: The first sentence says that "understanding"…"is rarely fully understood". Please rephrase.

Page1, Line 12: 12…166 → 12-166

Page1, Line 15: finding are → finding is

Page1, Clarify whether forward and backward are your own definitions or from the cited papers.

Page 5, Line 15: How do you calculate the water balance error? What is the reference?

Page 6, Line 17: You say that you use Larsim to use areal estimates of P, but then in line 20 you say that P is the model forcing.

Page 7, Line 4: easy to → easy way to

Equation B3 should look like:

$$\int_{t_0}^{t^*} Pdt \left( \frac{S(t^*)}{S_{max}} \right)^{\beta} - \int_{t_0}^{t^*} \left[ \int_{t_0}^{t} Pdt \beta \left( \frac{S(t)}{S_{max}} \right)^{\beta-1} \frac{1}{S_{max}} \frac{dS}{dt} \right] dt \tag{1}$$

There are two main differences from your formulation.

First, $uv\big|_{to}^{t^*} = u(t^*)v(t^*) - u(t^0)v(t^0) \neq \left( u(t^*) - u(t^0) \right)\left( v(t^*) - v(t^0) \right)$

Note that $u(t^0)v(t^0) = 0$

Second, in the last term of the equation, you have taken $\int_{t_0}^{t} Pdt$ out of the integral. This cannot be done,

because $\int_{t_0}^{t} Pdt$ is a function of time.

I made a simple Matlab example (see below) to show the difference in results between Equation 1 and the one you have in your paper. To simplify the calculation, I assumed beta=1. You can see that Equation 1 gives the right results, returning cumsum(Q), whereas your equation provides a result that differs significantly from cumsum(Q).

The problem in Equation B3 propagates through the whole paper, and strongly affects your results.

I recommend checking this (and the rest of the equations) before I proceed in further reviewing the paper.

```matlab
clear all
close all
% Fabrizio Fenicia, review of Simon Seibert et al, hess-2016-109
% Matlab code

P=[0;2;5;1;0;20;0];
Smax=50;
% beta=1;

nT=length(P);
S=NaN(1,nT);
Q=NaN(1,nT);
delT=1;

Sst=20;
S(1)=Sst;

% storage and flux from HBV model, using fixed step implicit approximation
for i=1:nT
    S(i)=Smax*(delT*P(i)+S(i))/(delT*P(i)+Smax);
    Q(i)=P(i)*(S(i)/Smax);
    if i<nT
        S(i+1)=S(i);
    end
end

figure
t=1:nT;
hold on
plot(t,S,'r');
plot(t,Q,'b');

sumQ=sum(Q);

cumP=sum(P)*delT;

% My implementation
add1_a=cumP*S(nT)/Smax;
cumSumP=cumsum(P);
dSdT=S(2:end)-S(1:end-1);
dSdT=[dSdT 0];

add2_int=cumSumP(:).*dSdT(:)/Smax;
add2_a=sum(add2_int)*delT;

sumQ_est_a=add1_a-add2_a;
```

```matlab
res_a=sumQ_est_a-sumQ;

%%%%%%%%%%%%%%%%%%%%%%%%%%%%%

% Your implementation

add1_b=cumP*(S(nT)-Sst)/Smax;
add2_int=sum(dSdT(:)/Smax)*delT;
add2_b=cumP*add2_int;

sumQ_est_b=add1_b-add2_b;

res_b=sumQ_est_b-sumQ;
```

---

## Author Response (AR2)

Dear Editor,

we gratefully acknowledge your assessment of our derivation of Eq. 5. We revised Appendix B and Eq. 5 according to your suggestion. As Erwin Zehe pointed out, we were looking for a good theoretical justification to scale CumP(t) with Smax, beyond plausibility. If we agree on the usefulness of the beta store to conceptualize direct runoff formation, the correct derivation still justifies this scaling very well.

Next to the derivation of Eq. 5 we revised the article according to your suggestions and according to the comments provided by the two referees. Below we provide a point-by-point reply.

**Editor comments**

Page1, Line 1: The first sentence says that "understanding"..."is rarely fully understood". Please rephrase.
*Authors reply: We rephrased the first sentence entirely.*

Page1, Line 12: 12...166
*Authors reply: changed to 12-166*

Page1, Line 15: finding are → finding is
*Authors reply: Changed to findings are*

Page1, Clarify whether forward and backward are your own definitions or from the cited papers.
*Authors reply: Revised to: The latter includes what we call "forward" and "backward"*

Page 5, Line 15: How do you calculate the water balance error? What is the reference?
*Authors reply: rephrased to "...the total accumulated water balance error of the entire period of four hydrological years, considering simulated evaporation, was smaller than 5 % of total precipitation."*

Page 6, Line 17: You say that you use Larsim to use areal estimates of P, but then in line 20 you say that P is the model forcing.
*Authors reply: Sorry for being imprecise. We revised the paragraph, compare Page 6, Line 17-22.*

Page 7, Line 4: easy to --> easy way to
*Authors reply: Done*

Equation B3 should look different
*Authors reply: This is correct, we oversaw that cumP, though being the accumulated rainfall mass, is a function of the upper integral limit, in fact it is defined a parametertic integral. Thanks a lot for pointing this out. We revised the manuscript according to your suggestion.*

**Review 1**

The text is significantly revised (and shortened!), and the story is much clearer in this recent draft. However, there are points that could be improved upon through minor revision of the text and some tempering of conclusions drawn. One of my main major criticisms is that there is a lot described in the text in terms of comparisons ACROSS sites, but no cross-site figures. The authors may wish to consider removing Figure 7 and replacing with a cross-site comparison figure that highlights the differences between type I and type II curve indices, and to frame this in the context of the interpretation presented in the discussion.

*Authors reply: We thank the referee for his/her positive feedback. We followed your suggestion and replaced Figure 7 with a multi-panel cross-site figure to provide a visual comparison of the different signatures.*

**Major comments:**

-A significant portion of your results is the discussion of your regression between catchment characteristics and dDMC characteristics. Unless I missed it, I did not see any mention of thresholds for significance or inclusion/exclusion of important variables within the multiple regression. We know that many hydrological relationships are nonlinear, and so merely testing using a linear correlation coefficient could neglect other important relationships that may exist.

*Authors reply: You are right that the thresholds for significance were not provided in all cases. Sorry for that and thank you for pointing this out. We added information on the thresholds in the revised version of the manuscript (compare chapter 3.1.4). All of the reported statistical relations were highly significant (p-values <0.001). In the multiple regression analysis we tested all variables reported in Table 2 and 3 plus those mentioned in the text. We kept those variables that increased the r2 by at least 5 % and which had p-values <= 0.05. We did however not test nonlinear relationships as clear hypotheses on the kind of relationships were missing. We hence assumed that linear relationships would suffice to identify important explanatory variables. We added this explanation to the manuscript, see P11 lines 11-12.*

- Are you suggesting that metrics related to dDMC be used in addition to SFDC, or are a better approach than SFDC to characterize rainfall-runoff relationships?

*Authors reply:  Better (or worse) are difficult terms. We propose that dDMCs provide means to characterize seasonal rainfall-runoff regimes that can not be extracted from other signatures such as FDCs which are frequently used to characterize similarity. This holds true even if FDCs are based on seasonal data. dDMCs provide different and hence additional information.*

-I am not convinced that an r2 of 0.22 indicates that two variables interact to influence tau (pg 14 lines 5 – 15). An r2 of 0.22 is still a very poor value. I would consider tempering some of the statements made about relationships between landscape characteristics and response variables. Again, are these relationships statistically significant? If anything, these results indicate that there are probably more disparate controls related to landscape

variables not included here or competing effects between landscape variables and climate that mediate many of these relationships.

*Authors reply: We agree that an r2 of 0.22 is a poor value. However, we disagree that the finding is not important. This is because a) the relationship between gradient, flux and resistance is physically out of question and thus meaningfully and b) the result is statistically significant. To clarify the latter we added information on the p-values of the regression analysis. The product of topographic gradient and saturated hydraulic conductivity and winter runoff coefficient yields a p-value < 0.001 whereas the two predictors alone only have much smaller r2-values and p-values of 0.08 and 0.009 respectively (see chapter 3.1.4). Please specify the statements that are formulated too strong.*

-There were many grammatical issues and missing words throughout. I suggest the authors give the manuscript a careful and thorough read.

*Authors reply: We apologize for that and assure proof reading of the manuscript by a native speaker.*

- Your scope neglects your comparison to SFDCs, which I see as a potential contribution to the manuscript, given how much you have highlighted it. Consider restructuring these questions to include reference to SFDC.

*Authors reply: We agree that we did not mention the comparison between DMC and FDC in the scope of the article. Thank you for pointing this out. We added a statement on the comparison between FDC and DMC after the research question Q3 (see P4 line 32): "Additionally we evaluate whether flow duration curves (FDC) might provide similar insights into seasonal runoff behavior as dDMC".*

-Figure 7: I find it difficult to connect the point you are trying to make with this figure, and I am not sure it is needed. What would be more useful would be to see comparison of type I and type II curve outputs, and how these could be interpreted to provide different information across your study catchments to augment the discussion on pages 17 and 18.

*Authors reply: The potential and limitation of the different dDMC types is discussed in detail in chapter 3.1, 4.1 and 4.2. Furthermore, a visual comparison between the two approaches is provided by the 6-panel figures 4 and 5. We however considered your suggestion and revised section 3.2. This includes the replacement of figure 7 by a multi panel plot that shows maps of the different signatures, i.e. the flow duration curve and properties of type I and type II dDMCs. This highlights differences in the spatial patterns. We conclude that each of the approaches reveals different patterns and hence, different information.*

**Minor comments:**

Pg 1 line 3: confusingly written

*Authors reply: We don't see any problem with Pg 1 line 3: "Specifically, we present dimensionless double mass curves (dDMCs) which allow studying runoff formation and the water balance at the seasonal and annual time scale." What do you mean?*

Pg 1 line 8: grammar
*Authors reply: Same thing, where is the mistake in "We show that different references result in different diagnostics. As such we define two kinds of dDMCs which allow to …"*

Pg 1 lines 16 – 18: vegetation vs. winter period – perhaps there are more scientific options for these terms? Growing season?
*Authors reply: We changed vegetation period to growing season.*

Pg 2 line 3: incorrect citing
*Authors reply: We corrected the citation.*

Pg 4 lines 9 through 30 I found to be confusing. The postulating, followed by Q1, make it difficult for the reader to follow your logic without having already read the paper. After reading the paper, I understand the points you are trying to make, but I found a lot of this to be almost wishy-washy. If it were interspersed with each of the questions to perhaps motivate these different investigations (and why you are doing them), I think the reader might draw more from what you have written.
*Authors reply: We revised the paragraph and hope that the new version is less confusing.*

Pg 11 Line 6: reference to incorrect section?
*Authors reply: We updated the cross-reference*

Pg 13 line 26 Reference tables here? There were more than 24 characteristics in Table 2 and 3 – it's worth noting which were included in the regression analysis
*Authors reply: We added a reference to Table 2 and 3 and clarified that all variables were included in the regression analysis.*

**Reviewer II: rejected**

This is the second time to review the manuscript. Compared to the first draft, the authors' focus has been clarified; on the effectiveness of dimensionless double mass curves (dDMC). However I have still significant difficulty in understanding the novel contribution of this entire manuscript and obtaining the clear logic in this manuscript. Essentially the authors propose to normalize the double mass curves by total precipitation or storage, which can be fairly easily done. Then a number of different hydrological or catchment properties were compared. Finally the authors state the dDMC is meaningful signatures for catchment runoff formation but without logical reasoning.

**Authors reply:** We regret that referee #2 does not understand the purpose and logic of our contribution. Essentially we propose that dDCMs are suitable signatures for characterizing rainfall-runoff dynamics at the seasonal time scale. dDMCs provide information that cannot be distilled from other signatures such as flow duration curves. The latter are widely used to describe similarity among catchments. In our manuscript we derive different kind of dDMCs and discuss their potential and limitations.

[revised manuscript text omitted]